# Projected soil carbon loss with warming in constrained Earth system models

Shuai Ren [1,2], Tao Wang [1] ✉, Bertrand Guenet[3], Dan Liu[1], Yingfang Cao[1,2], Jinzhi Ding [1], Pete Smith [4] & Shilong Piao [1,5]

The soil carbon-climate feedback is currently the least constrained component of global warming projections, and the major source of uncertainties stems from a poor understanding of soil carbon turnover processes. Here, we assemble data from long-term temperature-controlled soil incubation studies to show that the arctic and boreal region has the shortest intrinsic soil carbon turnover time while tropical forests have the longest one, and current Earth system models overestimate intrinsic turnover time by 30 percent across active, slow and passive carbon pools. Our constraint suggests that the global soils will switch from carbon sink to source, with a loss of 0.22–0.53 petagrams of carbon per year until the end of this century from strong mitigation to worst emission scenarios, suggesting that global soils will provide a strong positive carbon feedback on warming. Such a reversal of global soil carbon balance would lead to a reduction of 66% and 15% in the current estimated remaining carbon budget for limiting global warming well below 1.5 °C and 2 °C, respectively, rendering climate mitigation much more difficult.

Soil is the largest reservoir of terrestrial organic carbon[1], and there is compelling experimental evidence of accelerated soil carbon loss with warming[2,3], suggesting that soils may act as a positive carbon feedback on climate change. However, it is still not clear how global warming will affect soil carbon dynamics[4,5], either in terms of the magnitude of the effect or even its sign. One significant, and poorly understood, component of the system is soil carbon turnover[1,6], which is defined as the average time it takes for a carbon atom to enter and leave the soil system[7]. In the sub-models of Earth system models (ESMs), the soil organic carbon is generally viewed as a heterogeneous mix of two to several pools with different degrees of decomposability[8], and the soil carbon decomposition is commonly dictated by intrinsic decay constants, which are modified only by abiotic factors (temperature and moisture) with few spatially-uniform parameters[9–12]. The models vary greatly in the assumed values of constants and parameters[12], and the observational constraints on these values are generally lacking[1,4,13]. This

deficiency in model structure and/or parameters is manifested in simulations of soil carbon turnover time at local to regional scales which diverge both from each other[12,14] and from radiocarbon-based observations[6,15], reducing confidence in forecasts of how soil carbon stock will respond in the long term.

Here, we take advantage of the growing number of long-term soil incubation experiments to generate a spatially-explicit understanding of intrinsic soil carbon turnover ($\tau_i$) across the globe[16,17]. Notably, soil $\tau_i$ is representative of the theoretical carbon turnover time under optimal conditions. While various environmental constraints such as freezing and physical protection could inhibit the achievement of this theoretical value, leading to longer apparent value of $\tau_i$ in the real-world settings[8] (Methods). Using global soil $\tau_i$ observations, we then develop an observationally-calibrated three-pool model, which includes emerging concepts of controls on soil carbon stabilization (e.g., soil physical-chemical protection and priming effect)[18–21], to

---

[1]State Key Laboratory of Tibetan Plateau Earth System, Environment and Resources (TPESER), Institute of Tibetan Plateau Research, Chinese Academy of Sciences, Beijing, China. [2]University of Chinese Academy of Sciences, Beijing, China. [3]Laboratoire de Géologie, École normale supérieure, CNRS, PSL University, IPSL, Paris, France. [4]Institute of Biological and Environmental Sciences, School of Biological Sciences, University of Aberdeen, Aberdeen AB24 3UU, UK. [5]Institute of Carbon Neutrality, Sino-French Institute for Earth System Science, College of Urban and Environmental Sciences, Peking University, Beijing, China. ✉e-mail: twang@itpcas.ac.cn

constrain projected soil carbon changes in ESMs by the end of this century under different emissions scenarios.

## Results and discussion

We assessed the soil carbon turnover time by assembling a global database of aerobic carbon dioxide ($CO_2$) efflux data from temperature-controlled soil incubation studies conducted at 178 sites covering eight biomes, ranging from Arctic permafrost to dry Mediterranean forests (Fig. 1a). Due to the large heterogeneity of soil carbon, we represent the time evolution of soil $CO_2$ efflux ($SR$) using a three-pool model which partitions the total soil carbon ($C_{tot}$) into three reservoirs, with each decaying at its own intrinsic turnover rate (that is, the inverse of turnover time $\tau$)[16,17].

$$SR = \sum_{p=1}^{3} \frac{1}{\tau_p} \cdot C_{tot} \cdot f_p \qquad (1)$$

where $f$ is the partitioning coefficient of $C_{tot}$ for each pool ($p$). Using deconvolution analysis, the pool-specific $\tau$ values were quantified at their own incubation temperatures, and then scaled to a common temperature of 15 °C (ref. 22; see Methods). Note that the soils were incubated at constant temperatures, while other environmental factors such as soil moisture were generally maintained at the optimum level[23], suggesting that the inverted carbon pool-specific values of $\tau$ can be considered as intrinsic ones without environmental constraints.

### Global patterns of intrinsic soil carbon turnover times

There were considerable variations in soil $\tau_i$ within each biome and across biomes for each of the three carbon reservoirs (Fig. 1b–d). We

then used a boosted regression trees (BRT) model to determine the dominant environmental drivers (e.g., vegetation growth, local climate and soil attributes consisting of bulk density, pH value, organic carbon, total nitrogen, C:N ratio and soil texture) of the cross-site $\tau_i$ variations. Of the tested predictors, mean annual temperature (MAT) was the most important variable in explaining cross-site variability of $\tau_i$ for all three soil carbon pools, with importance values of 27–42% across the different pools (Supplementary Fig. 1). The normalized difference vegetation index (NDVI) and soil organic carbon came second in explaining $\tau_i$ variability in the fast soil carbon pool ($C_{fast}$) and in the slow ($C_{slow}$) and passive ($C_{passive}$) carbon pools. Specifically, soil $\tau_i$ had positive correlations with MAT (Supplementary Figs. 2–3), with lower MAT being associated with shorter $\tau_i$.

Based on the empirical relationships between environmental predictors and soil $\tau_i$, we generated a predictive model that could explain more than 75% of the cross-site variability in $\tau_i$ of each soil carbon pool without bias (Supplementary Fig. 1). This predictive model enabled us to interpolate $\tau_i$ of each soil carbon pool to a depth of 1 m across global soils (Fig. 2a, d, g). We derived a global $\tau_i$ of 0.3 yr for $C_{fast}$, with a spatial variation ranging between 0.06 yr (first percentile) and 0.64 yr (99th percentile) (Fig. 2a and Supplementary Table 1). The global $C_{slow}$ and $C_{passive}$ $\tau_i$ are 6.68 (2.3–12.1) and 398 (89–696) yr, respectively, with the values in parentheses denoting the spatial range between the first and 99th percentiles (Fig. 2d, g and Supplementary Table 1). We also calculated carbon-weighted $\tau_i$ to a depth of 1 m using the fraction of each soil carbon pool. The fraction of each soil carbon pool at the global scale was extrapolated from that derived at site level, using the empirical relationships between fraction of specific pools and environmental drivers across sites (see Methods). The global carbon-weighted $\tau_i$ is 316 yr with a spatial variation ranging between

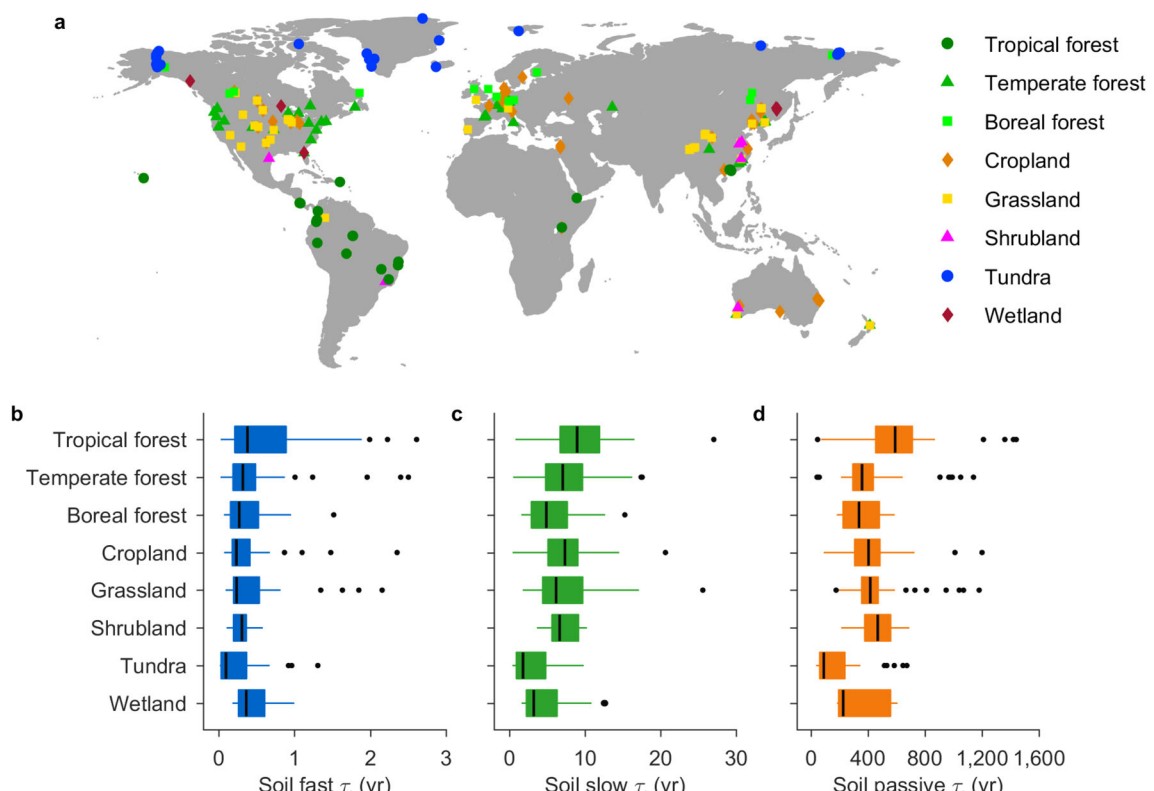

**Fig. 1 | Distribution of sample locations and intrinsic soil carbon turnover data.** **a** The spatial distribution of long-term temperature-controlled soil incubation experiments. **b–d** The boxplots showing the distributions of intrinsic soil carbon turnover times ($\tau_i$) of $C_{fast}$ (**b**), $C_{slow}$ (**c**) and $C_{passive}$ (**d**) that are inverted from all experiments within each of the eight biomes, respectively. The eight biomes are tropical forest ($n = 53$), temperate forest ($n = 73$), boreal forest ($n = 33$), cropland ($n = 59$), grassland ($n = 69$), shrubland ($n = 8$), tundra ($n = 64$) and wetland ($n = 15$), respectively. On each box, the central black line marks the median, the edges of the box correspond to the 25th and 75th percentiles, the whiskers extend to the range of the data, and the outliers are shown as dots.

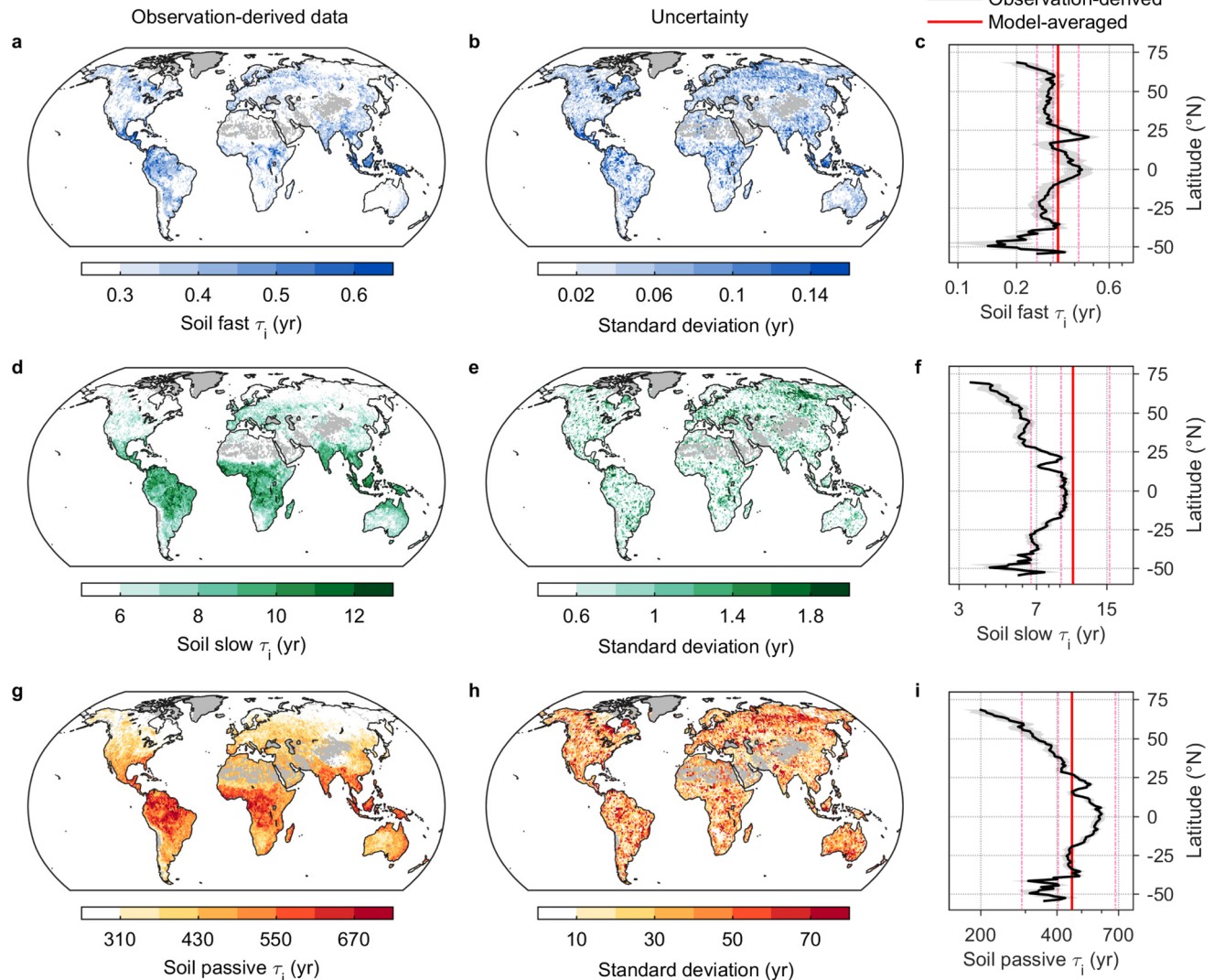

**Fig. 2 | Global distributions of intrinsic soil carbon turnover times.**
**a, d, g** Observation-derived intrinsic soil carbon turnover times ($\tau_i$) at 1 m depth that are upscaled from 374 data points at 15 °C using the machine learning model for $C_{fast}$ (**a**), $C_{slow}$ (**d**) and $C_{passive}$ (**g**), respectively. **b, e, h** Standard deviations of global soil $\tau_i$ estimates derived from different combinations of data sets. **c, f, i** Latitudinal profiles of soil $\tau_i$, aggregated at 0.5° latitudinal resolution. The black and red lines indicate observation-derived and ESMs-averaged soil $\tau_i$, respectively. The pink dashed lines are soil $\tau_i$ specified in each model (Supplementary Table 2). The red shading represents the standard deviation. The x-axis data in **c, f, i** are log-transformed.

81 yr (first percentile) and 609 yr (99th percentile) (Supplementary Table 1).

The longest $\tau_i$ values were found in tropical forests (0.43, 9.41 and 601 yr for $C_{fast}$, $C_{slow}$ and $C_{passive}$, respectively), shrublands (0.31, 7.8 and 467 yr) and grasslands (0.28, 6.93, and 418 yr), whereas tundra (0.25, 4.48 and 288 yr) and boreal forests (0.32, 5.58, and 326 yr) have the shortest values (Supplementary Table 1). We also find that the $\tau_i$ values of the three carbon pools vary considerably with latitude, with higher values in the tropical zone between 20° N and 20° S (mean $\tau_i$ is 0.35, 9.1 and 543 yr) but lower values in the northern high latitudes (0.27, 4.99 and 269 yr above 50° N) (Fig. 2a, d, g). To explore the extent to which the intrinsic turnover times translate into the actual ones due to environmental constraints[8], we compared our estimates with the radiocarbon-derived carbon age as a surrogate of realized or apparent turnover times (Supplementary Table 1). We find that soil $\tau_i$ is more than 16 times shorter than the actual one across the globe (316 yr versus 5,238 yr), and this value is largest in tundra (58.6) and boreal forests (18) but smallest in tropical forests (4.4) (Supplementary Table 1; Supplementary Fig. 4).

## Observational constraints on projected soil carbon dynamics in ESMs

Next, we evaluated soil $\tau_i$ specified in the five ESMs which have three soil carbon pools ($C_{fast}$, $C_{slow}$ and $C_{passive}$) archived in the Coupled Model Intercomparison Project Phase 6 (CMIP6). Soil $\tau_i$ in ESMs are generally described as global constants without any spatial variability (Supplementary Table 2). We found that ESMs overestimated soil $\tau_i$ in high-latitude ecosystems for all carbon pools (Fig. 2 and Supplementary Fig. 5), with a factor of 1.2, 2 and 1.7 for $C_{fast}$, $C_{slow}$ and $C_{passive}$, respectively. By contrast, in the tropics, soil $\tau_i$ are overestimated to a less extent, and even underestimated in some regions (Supplementary Fig. 5). Overall, the model ensemble overestimated soil $\tau_i$ by about 30% across the three different pools globally, with larger biases in $C_{slow}$ (58%) than those in $C_{fast}$ (12%) and $C_{passive}$ (17%) (Fig. 2). The model-data bias of soil $\tau_i$ could be attributed to the omission of critical microbial processes from ESMs, such as thermal adaptation[24]. Specifically, microbial turnover rates have been shown to adjust to temperature changes via biochemical trade-offs in enzyme and cell membrane structure and function[25,26]. Low temperatures typically select for

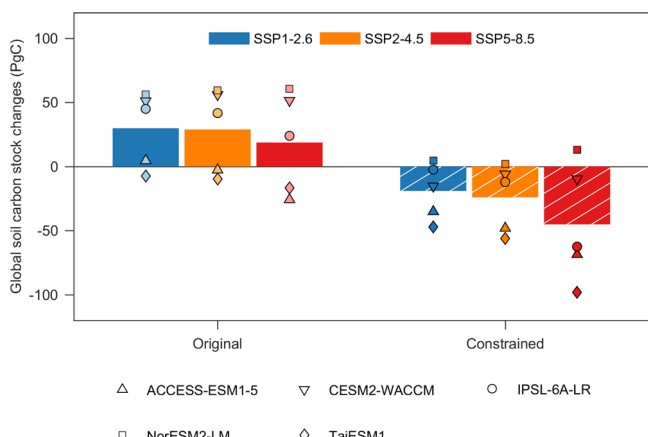

**Fig. 3 | Projected changes in global soil carbon stock.** Changes in global soil carbon stock between the current period (2005–2014) and the end of the century (2080–2099) from the original and constrained CMIP6 models under SSP1-2.6 (blue), SSP2-4.5 (orange) and SSP5-8.5 (red) emissions scenarios, respectively.

enzymes and/or membranes that are highly flexible to efficiently alter conformation and facilitate interactions[27]. As a result, cold-adapted microbial communities have faster growth and respiration rates than the warm-adapted when compared at common temperatures[5,26]. Future modeling efforts should seek a spatial representation of soil intrinsic turnover parameters especially for $C_{slow}$, e.g., by incorporating microbial metrics (such as thermal adaptation[24], species composition[28]) into ESMs to build confidence in predicting soil carbon-climate feedback.

The large deviation of modeled $\tau_i$ from the observations could lead to substantial biases in projections of soil carbon stock changes. To remove simulation biases due to misinterpretations of $\tau_i$ in ESMs, we firstly developed a reduced-complexity three-pool model, which could partition soil organic carbon into components with different intrinsic turnover rates, to emulate soil carbon simulations in much more complex ESMs (see Methods). Although our constructed reduced-complexity model is capable of mimicking ESMs soil carbon dynamics and making projections (Supplementary Fig. 6), it is limited to rudimentary processes without integrating emerging knowledge of controls on soil carbon turnover time. To improve the realism of the model, we constructed a refined reduced-complexity model to constrain ESM projections under three different emissions scenarios (SSP1-2.6, SSP2-4.5 and SSP5-8.5; Supplementary Fig. 7). In the refined reduced-complexity model, we described the soil carbon decay as a function of $\tau_i$ that is not only modified by climate factors, but also by the two competing processes (mineral protection and the rhizosphere priming effect) (see Methods). We forced the model with climatological mean (2000–2014) climate variables and satellite-based net primary productivity (NPP), and calibrated the model against both the gridded historical soil carbon pool and upscaled field determinations of soil carbon age (as a proxy of apparent soil carbon turnover times) derived from radiocarbon measurements[15] to minimize the model-data mismatch for the historical period (see Methods). The parameter uncertainties were obtained from different combinations of soil organic carbon (SoilGrids and the Harmonized World Soil Database) and NPP data sets (MODIS and Global Inventory Modeling and Mapping Studies) (Supplementary Table 3 and Supplementary Fig. 8).

The refined reduced-complexity model, constructed in this way, was run forward to 2100 using bias-corrected NPP and climate forcing (temperature and precipitation) under three different emissions scenarios (see Methods). An ensemble of original ESMs projected an increase of 18.6–29.8 petagrams of carbon (PgC) in global soils by the end of this century relative to the period 2005–2014 across the three scenarios (Fig. 3 and Supplementary Fig. 9). In contrast, constrained

ESM simulations using the refined reduced-complexity model showed that global soils will lose 19.1 Pg of carbon under SSP1-2.6, and this loss will be further escalated to 24 and 45.2 PgC under SSP2-4.5 and SSP5-8.5, respectively (Fig. 3). These projections translate into an estimated annual soil carbon loss rate of 0.22–0.53 PgC from now until the end of this century under different emissions scenarios. This finding is generally consistent with the expectation that the intrinsic turnover time was overestimated in complex ESMs (Fig. 2). This is because faster intrinsic soil carbon turnover would lead to a more rapid decline in soil carbon stock in response to warming, although environmental constraints such as climatic limitation and physical protection would inhibit the achievement of this intrinsic value[8]. Such soil carbon losses could largely offset increases in plant biomass due to the fertilization effect of rising $CO_2$, thereby reducing the potential capacity of land biosphere carbon sequestration in future. For example, in an intermediate emission scenario, the increase in $CO_2$ expected by 2100 would enhance the global plant biomass by 59 PgC using an empirical upscaling of $CO_2$ fertilization effect[29]. However, this enhancement in biomass carbon stock would be substantially counterbalanced by soil carbon losses (24 PgC).

In most regions, the soil carbon stock switched from a sink in the original simulations to a source in the constrained simulations (Fig. 3). Such a reversal of the soil carbon balance is most prominent in the arctic tundra and boreal regions (Supplementary Fig. 10), and occurred there even without invoking warming-induced permafrost soil carbon degradation[30,31]. The arctic tundra and boreal region have the shortest soil-carbon intrinsic turnover time of all the regions (Fig. 2), and the release of environmental constraints *via* warming and the plant rhizosphere priming effect will largely decrease the apparent soil carbon turnover time, thereby inducing large soil carbon losses. For example, a recent synthesis showed that rhizosphere priming could induce an additional 40 PgC loss from northern soils by 2100[32]. Furthermore, the entire tropical forest soils, including that in Amazonia, will also become a net carbon source of 8.2–21.4 PgC (Supplementary Fig. 10). Our results are in broad agreement with previous in situ experiments which showed that warming could accelerate the rate of soil carbon losses in tropical forests[3,33]. However, the magnitude of soil carbon losses from tropical forests under SSP5-8.5 (21.4 PgC) is lower than a recent estimate (65 PgC) that is simply extrapolated from a two-year warming experiment in a tropical forest under 4 °C warming[3]. The relatively low soil carbon loss is also due to the large increase in soil carbon input due to $CO_2$ fertilization in ESMs. While emerging evidence suggests that this $CO_2$ fertilization effect in ESMs may have been overestimated[34], and our estimates of soil carbon losses are then likely underestimated.

## Implications for the global remaining carbon budget

The vanished capacity of soils to sequester $CO_2$ suggests that a more aggressive strategy toward emissions reduction is required to realize the pledges of the Paris Agreement. A likely (50%) chance of keeping warming well below the 1.5 °C and 2 °C temperature targets requires that the maximum permitted carbon emissions remain below 68 PgC and 327 PgC from the start of 2022[35], respectively. We further estimated the amount of carbon absorbed by global soils to be less than the original ESM projections at the end of this century (Supplementary Fig. 11). The results imply that the currently estimated remaining carbon budget should be reduced by nearly 15–66% over the course of this century to achieve the warming goals, rendering climate mitigation much more difficult.

In summary, we synthesized long-term incubated-soil $CO_2$ flux measurements to provide observationally-constrained estimates of future soil carbon stock changes, using an optimized three-pool soil carbon model which includes the emerging concept of controls on soil carbon turnover time. Our observational constraints supported the current idea that global soils will create a strong positive carbon

feedback to climate change in a warmer world[4,5,36], and indicated that such soil carbon-climate feedback is substantially underestimated in current ESMs due to poor parameterizations of soil carbon turnover processes. The observationally-calibrated soil carbon model, with its spatially-varying estimates of parameters, is effective at capturing soil carbon turnover across diverse terrestrial biomes. It indicates the areas which require improvement in conventional models which parameterize carbon turnover processes in a spatially-uniform manner. The study provides important insights into the potential use of an observationally-constrained soil carbon model on projected soil carbon-climate feedback. However, our constrained projection is still subject to uncertainties, due to the omission of deep permafrost carbon dynamics[11], a divergent projection of soil carbon inputs[37], and a lack of feedbacks between soil carbon dynamics and nutrients. By including more data, particularly from under-sampled regions, such as Africa, central and southern Asia and some high latitudes, similar constrained projection studies are likely to provide further value to this area of research.

## Methods

### Collection of long-term soil incubation experiments

The intrinsic soil carbon turnover time reflects kinetic properties of various soil organic compounds under optimal conditions[8]. Generally, molecular structures with greater complexity exhibit an increased resistance to decomposition[38], resulting in a longer $\tau_i$. However, in real-world conditions, various environmental constraints, including freezing, flooding and physical protection, can dampen decomposition processes[8], frequently leading to a longer apparent value of $\tau_i$. Laboratory soil incubation experiments, with incubation duration varying from days to years, are a widely-used approach to estimate the intrinsic decomposability of soil carbon pools by measuring soil $CO_2$ fluxes under controlled conditions[17,39]. Results from such soil incubation experiments are invaluable for informing process-based ecosystem models about carbon pool sizes and their intrinsic turnover rates[40]. In contrast to short-term incubations (a few days to weeks), where measured soil $CO_2$ fluxes mostly originate from respiration in the fast-cycling carbon pool, long-term incubations can provide more information on the decomposability of slow-cycling carbon pools[41,42].

Here, we assembled a raw dataset of soil $CO_2$ fluxes from soil incubation experiments to assess the intrinsic decomposability of different soil carbon pools. We included only soil incubations which satisfied the following criteria: (1) the length of the incubation experiment was longer than 6 months; (2) the aerobic $CO_2$ production must have been measured over the parallel-warming incubation experiments. If more than one incubated temperature was available, we selected the high-end temperature treatment, as a higher incubation temperature generally leads to a quicker depletion of the fast-cycling carbon pool and therefore enables more information on the slow-cycling carbon decomposability to be retrieved[17]; (3) the initial soil carbon concentration must be available; (4) the soils were incubated without any substrate addition; (5) soil respiration rates in the initial phase must be higher than those at the end of incubation; and 6) data can be taken directly from tables or extracted from figures using the GetData (v.2.25) software. In total, we gathered 102 peer-reviewed publications covering 178 sites (Fig. 1), resulting in a total of >5000 time-series of soil $CO_2$ flux data. The soil $CO_2$ flux data were taken from a range of depths from 5 cm to >1 m, with incubation temperatures ranging from 4 to 35 °C and incubation durations ranging from 180 d to >10 yr. The metadata for each sampling site included location (latitude and longitude), climate (mean annual temperature and precipitation), soil physicochemical attributes (bulk density, pH, soil organic carbon, total nitrogen, C:N ratio, and sand and silt content), vegetation types, sampling depth, incubation temperatures and duration. The samples from which data were obtained can be classified into tropical forest (23° S–23° N; $n = 53$), temperate forest (23–50° N

and S; $n = 73$), boreal forest ( > 50° N; $n = 33$), grassland ($n = 69$), cropland ($n = 59$), shrubland ($n = 8$), tundra ($n = 64$) and wetland ($n = 15$) (Fig. 1). For sampling sites where climate and soil attributes were not available, we extracted data from WorldClim version 2.0[43] and SoilGrids[44], respectively, based on the published location and depth information. The normalized difference vegetation index from the MODIS was also obtained for each sampling site as a measure of site productivity.

### Inversion of intrinsic carbon turnover times from soil incubation experiments

We assumed that the total $CO_2$ flux is composed of contributions from the three different carbon pools (that is $C_{fast}$, $C_{slow}$ and $C_{passive}$). For each sampling site, we fitted the time series of total soil $CO_2$ fluxes using the three-pool carbon decomposition model[17] described below. In this model, the total soil $CO_2$ flux ($R$, in mg C g⁻¹dw d⁻¹) was the sum of the respiration rates ($r$) derived from the three carbon pools ($p$) with different sizes and turnover rates (Eq. (2)).

$$R = \sum_{p=1}^{3} r_p = \sum_{p=1}^{3} k_p \times C_{tot} \times f_p \quad (2)$$

$$f_p = \frac{C_p}{C_{tot}}, \sum_{p=1}^{3} f_p = 1 \quad (3)$$

where the pool-specific respiration rate ($r_p$) was computed as the product of intrinsic pool-specific decay rate ($k_p$, in d⁻¹; the inverse of $\tau_i$), the total initial carbon pool ($C_{tot}$, in mg C g⁻¹dw) and a partitioning coefficient ($f_p$). The partitioning coefficient describes the ratio of carbon pool $p$ to the total carbon pool, with the sum of the three coefficients equal to unity (Eq. (3)).

The model parameters (**m**), consisting of $k$ and $f$ for each pool, were optimized against measured soil $CO_2$ fluxes (**O**) using a Bayesian probabilistic inversion approach[45]. This approach states that the posterior probability density function (PDF) of the model parameters $p(\mathbf{m}|\mathbf{O})$, conditional on the data (**O**), can be obtained by applying Bayes theorem[46].

$$p(\mathbf{m}|\mathbf{O}) \propto p(\mathbf{O}|\mathbf{m})p(\mathbf{m}) \quad (4)$$

where $p(\mathbf{m})$ is the prior PDF and $p(\mathbf{O}|\mathbf{m})$ is the PDF of **O** conditional on **m**, also called the likelihood function. For constructing $p(\mathbf{m})$, we first specified the range of each parameter according to values obtained from the literature (Supplementary Table 4) and then assumed that they are uniformly distributed over this range. The likelihood function $p(\mathbf{O}|\mathbf{m})$ summarizes the difference between the simulated and measured $CO_2$ flux data. Through assuming that the errors of measured $CO_2$ flux data follow a normal distribution with zero mean, the likelihood function $p(\mathbf{O}|\mathbf{m})$ is given by,

$$p(\mathbf{O}|\mathbf{m}) \propto \exp\left\{ -\frac{1}{2\sigma^2} \sum_{t \in obs(S_i)} \left[ S_i(t) - O_i(t) \right]^2 \right\} \quad (5)$$

where $O_i(t)$ and $S_i(t)$ are the measured and simulated $CO_2$ fluxes, and $\sigma$ is the standard deviation of the measured $CO_2$ fluxes.

To derive the posterior distribution of model parameters analytically, we resorted to the Metropolis–Hastings (M–H) algorithm, a powerful Markov Chain Monte Carlo (MCMC) technique for simulating complex and nonstandard multivariant distributions, which would iteratively search for the optimum feasible solution[47,48]. During the inversion process, the sum of the fractions ($f$) of all three carbon pools must be equal to unity, and the intrinsic decay rate ($k$) should be largest in the fast carbon pool and smallest in the passive one. We then

computed the maximum likelihood estimates for well-constrained parameters (e.g., the decay rates from the fast and slow carbon pools), the means of the poorly-constrained parameters (e.g., the decay rate from the passive pool), and the confidence intervals for all parameters from their posterior distributions. Overall, the performance of the three-pool model in fitting the measured soil $CO_2$ fluxes was good (Supplementary Figs. 12 and 13).

In the compiled data, soil incubation temperatures ranged from 4 to 35 °C, making it difficult to compare $\tau_i$ across studies in a quantitative manner. Therefore, we adjusted $\tau_i$ at different incubation temperatures to a reference temperature of 15 °C by using the following equation[22,49].

$$\tau_{i,15} = \tau_{i,T} \times Q_{10}^{\frac{T-15}{10}} \quad (6)$$

where $\tau_{i,15}$ and $\tau_{i,T}$ are soil intrinsic turnover times at the reference temperature of 15 °C and the incubation temperature T (°C), respectively. $Q_{10}$ is a temperature sensitivity parameter defined as the factor by which soil respiration increases with a 10 °C increase in temperature. Here, we assumed that $Q_{10}$ varies with temperature, and their empirical function was derived from a previous synthesis analysis of $Q_{10}$ and temperature from laboratory studies across various ecosystems at the global scale[23] (Supplementary Fig. 14).

## Upscaling of site-level intrinsic carbon turnover times to the global level

We used boosted regression trees (BRT) to assess the relative importance of the independent variables, including local climate (mean annual temperature and mean annual precipitation (MAP)), vegetation productivity (NDVI) and edaphic properties (bulk density, pH value, organic carbon content, total nitrogen content, C:N ratio, silt and sand content), on spatial variability in $\tau_i$ for each carbon pool (at 15 °C; Supplementary Table 5). Note that we selected local climate variables because they can affect the structure of soil microbial communities and then the soil carbon decomposition[50]. BRT is an ensemble method that combines the strengths of regression trees (tree-based models that relate a predicant to predictors through recursive partitioning) and boosting algorithms (using large numbers of relatively simple tree models to give improved predictive performance)[51]. Our approach, which is superior to most traditional methods, can handle different types of predictor variables and interaction effects between predictors, and does not require data transformation or outlier elimination. We used a grid-search procedure, using ten-fold cross-validation, to select the best hypermeter combination of BRT modeling with the lowest cross-validation root mean square error (Supplementary Table 6). The BRT analysis was performed using the *gbm* and *caret* packages in R 4.0.5. The soil $\tau_i$ data were log10 transformed before starting the analysis.

The constructed BRT model was able to explain more than 75% of the variances in $\tau_i$ across different sites (Supplementary Fig. 1), and was subsequently used to produce global maps of $\tau_i$ for each pool at a spatial resolution of 0.1°. The uncertainties of these parameters were further generated by forcing the constructed BRT models with combinations of different sources of climate and edaphic data (Fig. 2). For climate predictors, we used mean annual temperature and mean annual precipitation from both the Climate Research Unit (CRU) version 4.01[52] and WorldClim version 2.0[43]; for edaphic factors (soil physicochemical properties), we considered both the Global Soil Dataset for Earth System Modeling (GSDE)[53] and the SoilGrids[44] data (Supplementary Table 5). We also tested the extent of our extrapolations and found that our soil samples spanned most environmental conditions around the globe (Supplementary Fig. 15). Despite this, certain regions (such as Africa, central and southern Asia and some high latitudes) are underrepresented by our samples (Fig. 1). Thus, more long-term soil incubations are urgently needed in these specific regions.

To calculate soil carbon-weighted $\tau_i$ ($\tau_{i,w}$ in Eq. (7)), we used the same method as used to obtain global maps of fractions in different soil carbon pools ($f_p$ in Eq. (7); Supplementary Fig. 16).

$$\tau_{i,w} = \sum_{p=1}^{3} \tau_{i,p} \times f_p \quad (7)$$

## Development of a reduced-complexity three-pool model

From the CMIP6 historical and future simulations for three different emissions scenarios (SSP1-2.6, SSP2-4.5 and SSP5-8.5), we selected only those ESMs which reported the total soil carbon stock partitioned into the three discrete soil carbon components: *cFast, cMedium* and *cSlow*. For ESMs with litter or woody debris carbon pools (*cLitter* and *cCwd*), we combined these pools with *cFast* to form the fast-cycling carbon pool[6]. Since ESMs do not provide depth information for soil carbon, we assumed that the carbon was stored within the top one meter of the soil[6,12]. The models used here are ACCESS-ESM1-5, CESM2-WACCM, IPSL-CM6A-LR, NorESM2-LM and TaiESM1 (Supplementary Table 7).

We developed a reduced-complexity three-pool model to approximate soil carbon simulations in each grid cell of the five ESMs. This three-pool model partitions soil organic carbon into components with different intrinsic turnover rates (Eqs. (8–10)).

$$\begin{cases} \frac{dC_f}{dt} = NPP(t) - k_f \times C_f(t) \\ \frac{dC_s}{dt} = k_f \times C_f(t) \times r_f - k_s \times C_s(t) \\ \frac{dC_p}{dt} = k_s \times C_s(t) \times r_s - k_p \times C_p(t) \end{cases} \quad (8)$$

where $r_f$ and $r_s$ are transfer coefficients for carbon flowing from fast to slow pools and from slow to passive pools, respectively. $C_f$, $C_s$, $C_p$ represent the soil carbon stock of the fast, slow and passive pools, respectively. $k_f$, $k_s$ and $k_p$ are the actual carbon decay rates (yr⁻¹) calculated using Eq. (9).

$$k = \frac{1}{\tau_a} = \frac{1}{\tau_i} \times F(T) \times F(P) \quad (9)$$

$$F(T) = Q_{10}^{\left(\frac{T-T_{ref}}{10}\right)}; F(P) = P^b \quad (10)$$

Where $\tau_a$ denotes the actual-, and $\tau_i$ the intrinsic, carbon turnover time (yr) at a reference temperature ($T_{ref}$). T and P are the ambient temperature (°C) and precipitation (mm), respectively. $F(T)$ is a $Q_{10}$-based standard exponential function to represent the temperature modifier of $\tau_i$. $F(P)$ represents a moisture modifier that increases with annual precipitation, normalized to maximal annual precipitation for each ESM, using an exponential function where $b$ is greater than zero[12].

In the three-pool model, five of eight parameters, including intrinsic soil carbon turnover time of each pool, $Q_{10}$ and associated $T_{ref}$ were from original ESMs (Supplementary Table 2). In particular, for ESMs with more than three carbon pools, we aggregated $\tau_i$ of *cLitte* and/or *cCwd* into that of *Cfast*. The transfer coefficients ($r_f$ and $r_s$) and the environmental dependency parameter ($b$) were diagnosed using a Bayesian global optimization algorithm in each grid cell[54], and their uniform priors are shown in Supplementary Table 8. Specifically, we initialized the reduced complexity model with pool-specific carbon stock from historical ESM simulations (2000–2014), and then used the variables *npp, tas* and *pr* from different emissions scenarios as inputs to force the reduced model from 2015 to 2100 for each ESM. The objective function was constructed based on the root mean square error between the actual and modeled soil carbon stock for each of the three carbon pools. The reduced-complexity model with these diagnosed parameters was found to reproduce ESM soil carbon

simulations very well (Supplementary Fig. 6), suggesting that the three-pool model was an almost perfect approximation of the ESMs.

Notably, the five models used in this study may not be representative of the broader CMIP6 ensemble because they draw heavily from the CENTURY model and then have similar structures (e.g., three different soil carbon pools)[55]. Other CMIP models derived from models other than CENTURY may behave differently and deserve further exploration.

### Constrained projections of soil carbon stock changes

The model-data comparison suggested that $\tau_i$ in ESMs has a large deviation from interpolated observations at the global scale (Fig. 2). Here, to remove soil carbon simulation biases due to misinterpretations of $\tau_i$ in ESMs, we constrained the ESM projections of soil carbon sink by 2100 (Supplementary Fig. 7). Our constructed reduced-complexity model was demonstrated to well mimic soil carbon dynamics in ESMs (Supplementary Fig. 6), but it generally assumes that decomposition rates are only constrained by temperature and moisture availability[9–11]. In fact, emerging processes, such as mineral protection and the rhizosphere priming effect, have the potential to affect soil carbon turnover time[8], but which were mostly absent in current ESMs. We, therefore, refined the reduced-complexity model by including the impacts of climatic factors and these emerging processes (Eqs. (11–12)) on soil carbon turnover time.

$$k = \frac{1}{\tau_i} \times F(T) \times F(P) \times F(M) \times F(RP) \tag{11}$$

Where $k$ is the actual soil carbon decay rate (yr$^{-1}$) and $\tau_i$ denotes the intrinsic carbon turnover time (yr) at a reference temperature of 15 °C. $F(T)$ and $F(P)$ are temperature and precipitation modifiers described by Eq. (10). Specifically, $Q_{10}$ in $F(T)$ was spatially-heterogeneous and pool-specific, rather than a constant as assumed in ESMs[56]. Furthermore, the $Q_{10}$ values of different carbon pools follow the carbon quality-temperature hypothesis[38], with higher values for more recalcitrant carbon pools. $F(M)$, a scalar representing the mineral protection of soil carbon, is described as a function of soil clay content (*clay* in Eq. (12); Supplementary Fig. 17a)[57].

$$\begin{cases} F(M) = 24.2 \times clay & clay < 0.033 \\ F(M) = -2.1 \times clay^2 + 6.2 \times clay + 0.6 & clay \geq 0.033 \end{cases} \tag{12}$$

$F(RP)$, a scalar representing the plant rhizosphere priming effect, is given as a function of root respiration ($R_{root}$)[32].

$$F(RP) = 1 \Big/ \left(1 + \frac{2.47 \times R_{root}}{13.01 + R_{root}}\right) \tag{13}$$

This Michaelis–Menten function is derived from a global meta-analysis, which showed a positive relationship between the rhizosphere priming effect and root respiration ($R_{root}$ in Eq. (13)) across all studies (Supplementary Fig. 17b). The root respiration was estimated to be around 7% of NPP[32]. Here, the rhizosphere priming effect was assumed to only occur in slow and passive pools with "older" soil organic carbon[18,58]. The rhizosphere priming effect could also increase the release of soil nutrients, which would, in turn, stimulate plant growth[59] and thereby create a positive feedback loop that further decreases soil carbon turnover time. On the other hand, enhanced plant growth due to the rhizosphere priming effect might partly offset soil carbon losses due to enhanced soil carbon turnover rates. The net effect due to this positive feedback on soil carbon stock changes might not be so large. This effect was not included in our refined reduced-complexity models due to a general absence of carbon-nitrogen coupling processes in most of ESMs. To fully resolve this question, we require next generation of Earth system models that explicitly

incorporate the rhizosphere priming effect within the coupled carbon-nitrogen cycle framework.

In the refined reduced-complexity model, the parameter $\tau_i$ of each pool was taken from our interpolated observations at the grid cell level (Supplementary Fig. 7). Other parameters, including precipitation scalar $b$, pool-specific temperature scalar $Q_{10}$, as well as the transfer coefficients $r_f$ and $r_s$, were optimized through minimizing errors between the observed and modeled carbon content of the three pools in each grid cell (Supplementary Table 9). These parameters were empirically calibrated so as to ensure that the input rate for each observed soil carbon pool was coupled to $\tau_i$ which is modified by climatic factors, mineral protection and rhizosphere priming. The observed soil carbon content for each pool at the grid cell scale was derived from gridded soil carbon data multiplied by our estimated carbon content fraction for each pool (Supplementary Fig. 16). We used data-driven estimates of soil carbon turnover times from global radiocarbon measurements[15] as a surrogate for observed $\tau_a$ at the grid scale. In the optimization process, the 15-year mean (2000–2014) of climate and satellite-based NPP were used as inputs. In addition, to account for uncertainties in these empirical parameters, we considered the four combinations of the two gridded soil carbon data sets (SoilGrids and the Harmonized World Soil Database[60]) and the two net primary productivity data sets (MODIS[61] and Global Inventory Modeling and Mapping Studies[37]) in the optimization algorithm (Supplementary Table 3).

To initialize the soil carbon pool for the start of the future simulations, the refined three-pool model, with parameters calibrated using mean climate and NPP from 2000–2014, was further run repeatedly, using yearly climate and satellite-based NPP from the period 2000–2014 as inputs, for more than 30 cycles until it reached a steady-state condition. The steady-state condition is defined as the 15-year mean difference between NPP and simulated carbon decomposition rate from the three pools being approximately equal to zero. The inclusion of such a steady-state run is to avoid introducing an artefact into the projection of future soil carbon change. After reaching the steady-state condition, the size of the global soil carbon pool reached $2433 \pm 325$ PgC over the four ensemble predictions: a value that is well within the range of observation-based global soil C stock estimates (~1500–3000 PgC)[62,63].

Following the steady-state run, we then ran the refined three-pool model forward to 2100 using bias-corrected climate (temperature and precipitation) and NPP data simulated by CMIP6 ESMs under three different emissions scenarios (Supplementary Fig. 18). We applied the following delta or change method[64,65] to correct biases in the future climate and NPP at the monthly timescale (Eq. (14)).

$$\begin{cases} T_{fut,cor} = T_{bas,obs} + (T_{fut,mod} - T_{bas,mod}) \\ P_{fut,cor} = P_{bas,obs} \times \left(\frac{P_{fut,mod}}{P_{bas,mod}}\right) \\ NPP_{fut,cor} = NPP_{bas,obs} + (NPP_{fut,mod} - NPP_{bas,mod}) \end{cases} \tag{14}$$

where $T_{fut,cor}$, $P_{fut,cor}$ and $NPP_{fut,cor}$ are bias-corrected temperature, precipitation and NPP projections for 2015–2100, respectively. $T_{bas,obs}$, $P_{bas,obs}$ and $NPP_{bas,obs}$ are CRU-derived temperature and precipitation, and satellite-derived NPP during the baseline period 2000–2014, respectively. $T_{bas,mod}$, $P_{bas,mod}$ and $NPP_{bas,mod}$ are ESM-simulated temperature, precipitation and NPP for the baseline period 2000–2014, respectively, and $T_{fut,mod}$, $P_{fut,mod}$ and $NPP_{fut,mod}$ are ESM-simulated temperature, precipitation and NPP for 2015–2100, respectively.

**Sensitivity analysis.** To assess the robustness of our constrained results, we conducted the following sensitivity experiments (SE) (Supplementary Fig. 19; Supplementary Table 10). First, we used the Arrhenius function instead of the empirical $Q_{10}$-temperature

relationship in our default simulation to scale site-level $\tau_i$ at their own incubation temperatures to that at 15 °C and then further obtained global estimates of soil $\tau_i$ in a machine learning algorithm linking these $\tau_i$ to environmental variables across sites. We then prescribed $\tau_i$ using these data-driven estimates based on the Arrhenius function in refined reduced-complexity model to constrain soil carbon projections (SE1). Second, the use of laboratory incubation experiments, albeit with the length of the period longer than six months, would still have uncertainties in the quantification of $\tau_i$ of slow-cycling carbon pool especially $C_{passive}$. In SE2, we prescribed $\tau_i$ of $C_{passive}$ as the ensemble mean of ESM's own values rather than the data-driven estimates in our default simulation (Fig. 2).

Third, in SE3-SE6, we used the reduced-complexity model (Eq. 8), which only considered climate controls on soil carbon turnover time and used model inputs such as NPP, MAT and MAP directly from the original ESMs, instead of the refined model (Eq. 11) to constrain soil carbon projections in ESMs. In order to evaluate the relative importance of soil $\tau_i$ and $Q_{10}$ in soil carbon projections, we performed the following SE3-SE5 tests. For SE3, we replaced soil $\tau_i$ with our data-driven estimates, and replaced $Q_{10}$ for each pool with that derived from the refined reduced-complexity model (Eq. 11), in which $Q_{10}$ for each pool was obtained through calibration against observed soil carbon stocks (Supplementary Table 9). For SE4, we replaced soil $\tau_i$ with our data-driven estimates and used ESM's own $Q_{10}$ and associated reference temperatures. For SE5, we used ESM's own soil $\tau_i$, but assigned $Q_{10}$ for each pool to the calibrated one. The magnitude of soil carbon loss in SE4 (19.5 PgC averaged across scenarios) is much closer to that in SE3 (26.3 PgC) than in SE5 (13 PgC), suggesting that soil $\tau_i$ is more important than $Q_{10}$ in determining projections of soil carbon dynamics. In addition, since the use of laboratory incubation experiments would have uncertainties in the quantification of $\tau_i$ of $C_{passive}$, for SE6, we used the ensemble mean of ESM's own soil $\tau_i$ of $C_{passive}$, but assigned $\tau_i$ of $C_{fast}$ and $C_{slow}$, and $Q_{10}$ to be the same with those in SE4. The results of SE6 showed that global soils would be a source of carbon to the atmosphere (10 PgC), albeit at a lower magnitude than SE4.

**The impact of changes in global soil carbon stock on the remaining carbon budget**
Changed soil carbon sequestration potential due to observational constraints could affect the remaining carbon budget for limiting global warming below 1.5 °C and 2 °C[35]. To estimate the global soil carbon sequestration potential under the warming targets, we relied on a strong linear relationship between the change in global air temperature and the observationally-constrained changes in global soil carbon stock between the end of this century and the historical period across the five CMIP6 models under the three emissions scenarios (Supplementary Fig. 11). The change in global air temperature was calculated using the 1850–1900 mean as a baseline, while the calculation of the change in soil carbon storage used the 2020 value as a baseline.

**Reporting summary**
Further information on research design is available in the Nature Portfolio Reporting Summary linked to this article.

## Data availability
The outputs of the Earth system models can be downloaded from the CMIP6 website (https://esgf-node.llnl.gov/projects/cmip6/). The WorldClim and CRU climate data are available at http://www.worldclim.com/version2 and https://crudata.uea.ac.uk/cru/data/hrg/, respectively. Soil physicochemical attributes of the GSDE and SoilGrids data sets can be obtained from http://globalchange.bnu.edu.cn/research/soilw and https://soilgrids.org/, respectively. The soil carbon content of HWSD can be obtained from http://www.fao.org/soils-portal/data-hub/soil-maps-and-databases/harmonized-world-soil-database-v12/en/. The global NPP databases of MODIS and GIMMS3g are available at http://files.ntsg.umt.edu/data/NTSG_Products/MOD17/ and https://wkolby.org/data-code/, respectively. The collected meta-data and gridded maps of soil $\tau_i$ and fractions of different carbon pools have been deposited in the Figshare data repository (https://doi.org/10.6084/m9.figshare.19641759.v1)[66]. Source data are provided with this paper.

## Code availability
Data analysis was carried out using R v.4.0.5 and MATLAB R2016a. The code used in this study is available at the Figshare data repository (https://doi.org/10.6084/m9.figshare.19641759.v1)[66].

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

## Acknowledgements

This study was supported by the National Natural Science Foundation of China (42230411), the NSFC project Basic Science Centre for Tibetan Plateau Earth System (41988101), the Second Tibetan Plateau Scientific Expedition and Research Program (2019QZKK0606), the Science and Technology Plan Project of Tibet Autonomous Region (XZ202201ZY0015G), and Innovation Program for Young Scholars of TPESER (TPESER-QNCX2022ZD-02). We also acknowledge the support of Kathmandu Center for Research and Education, Chinese Academy of Sciences—Tribhuvan University.

## Author contributions

T.W. designed the research; T.W. and S.R. wrote the paper; S.R. performed the data analysis; all authors contributed to the interpretation of the results and to the text.

## Competing interests

The authors declare no competing interests.
