## [Peer Review File · Nature Communications]

Projected soil carbon loss with warming in constrained Earth system modelsREVIEWER COMMENTS

Reviewer #1 (Remarks to the Author):

Review of “Projected soil carbon loss with warming in constrained Earth system models” (NCOMMS-23-13941-T) for Nature Communications

Paper summary

In this paper, the authors present a compilation of observational data of soil carbon turnover times in different biomes and climate zones. This analysis shows that the intrinsic turnover time (i.e. turnover if the soil were at optimum environmental conditions) is highest in cold / high latitude regions, and lowest at warmer low latitudes. The seeming discrepancy with observed slow carbon degradation in high latitudes soils is because the apparent turnover time differs from the intrinsic turnover time due to non-optimum environmental conditions (so if high-latitude soil were subjected to more optimal conditions, the apparent turnover time would converge with the intrinsic turnover time). This data is then used to develop a simple model of soil carbon pools to estimate global patterns of intrinsic turnover time, the results of which compare favourably to estimates derived from a reduced complexity model emulating the soil carbon dynamics of Earth System Models. However, while the spatial patterns align the ESMS substantially over-estimate the magnitudes of intrinsic turnover time. To show what impact this has on future soil carbon projections, another model is developed by calibrating the previous reduced complexity model with intrinsic turnover time observations and adding additional processes (mineral protection and rhizosphere priming) and then run to 2100 under low/medium/high emissions scenarios. This indicates soil carbon will switch from a carbon sink to source under all scenarios, offsetting some of the carbon sequestered in plant biomass and so reducing the carbon budget for keeping global warming below 2C.

General comments

In this review I am focusing on the analysis logic and Earth system implications rather than the methodological details, which I’m not so familiar with for this field but appears to be sound, supports the conclusions made, and is likely reproducible from the data provided (although uploading code to a repository e.g. github/zenodo as well would improve reproducibility further).

In general this is an interesting and useful paper with clear relevance to Earth system modellers and future climate policy, and should be published with some text revisions to clarify some things. It is well known that soil carbon dynamics is one of the most uncertain aspects of Earth system models, and this study helps to illustrate and quantify the gap between ESMS and observations, indicating where ESMS can improve in future. However, there’s quite a lot packed in to this paper (with results from multiple data analyses and models included), so there’s not a lot of space to explain things fully and put them in context for readers who are less familiar with soil carbon turnover and its implications. For example, I think the difference between the intrinsic and apparent turnover times should be unpacked and

explained more, as it's not necessarily obvious to non-specialist readers and is a crucial point for understanding the results of this paper. Additional explanation on model rationale and why ESMs are over-estimating turnover time and thus carbon sequestration would be helpful too, with the latter also important for outlining the implications of where ESMs could focus future improvements (which could be explicitly noted later too). Word count is no doubt a limitation here, but I think some brief extra sentences providing context where flagged may be sufficient to resolve this (otherwise a longer paper would be better to do this work justice). Finally, calculating the implications for the 1.5C carbon budget as well as 2C would no doubt be of interest to many. Specific details on where these points apply are below.

Specific comments

Line 23-27: Does "after observational constraints" mean from now, or later on? I suggest rephrasing this sentence a bit to make this clearer (could also consider splitting the sentence after "worst emission scenarios").

Line 29-29: As per the more detailed comment on this below, what about implications for the carbon budget for 1.5C?

Line 32: A minor point, but might be clearer language to say something like "may act as a positive feedback on climate change" rather than "would act..."

Line 33: I found "in the real Earth system" to be oddly phrased on first reading (as those experiments in the previous sentence were done on real parts of the Earth system too, albeit placed under artificial incubation conditions), though I do know that you mean across the whole Earth system in practice rather than in a few experiments here.

Line 35: I think it'd be useful either here or somewhere early on to briefly define and explain soil carbon turnover for less familiar readers, especially as this is not a disciplinary journal.

Line 47: what do you mean by a "compact" model? Either brief explanation of that later on, or if you mean reduced complexity, or simple, or stylised model then that that would be more familiar phrasing (at least to me).

Line 53: while the dataset gathered is pretty comprehensive across different biomes, there are some limitations (e.g. not so many datasets for non-forests, and not many sites in Africa or Central/South/Southeast Asia). I'm sure this reflects data availability and so is somewhat inevitable, but it could bear mentioning as a limitation in at least the methods section (as sometimes ecosystems behave a bit differently on different continents despite being in the same global biome). Also, in the map there are a few North European sites in e.g. the UK & North Germany that appear to be labelled as Boreal Forest when by most classifications they'd be in the Temperate biome (unless they're labelled as Boreal because they're from conifer plantations, but that would lead to different limitations) so this

might need checking, especially as this would reduce Palearctic Boreal representation. Lastly on Fig. 1, at first glance it'd be easy to assume the colours in panels b-d are meant to match the biome colours above, so for clarity might be preferable to make as different shades of a new colour if possible, or to maintain continuity with Fig. 2 re-do the 1a dots as different symbols instead (the red wetland dots in panel a may also clash with green for colour-blind readers).

Line 70: could briefly mention a headline quantification of the importance of MAT here from Fig S1 (looks like c. 27-42% across the different pools) to underline this point.

Line 73-77: the optimal environmental conditions caveat is key here, as a non-familiar reader might be otherwise surprised that colder regions are associated with faster soil carbon turnover given long-term C storage of say permafrost. I feel like this could be made clearer upfront to avoid confusion.

Line 100-103: again, I think this could do with highlighting / a bit more explanation to make it clear to less-familiar readers that intrinsic turnover time is what's expected in theory given otherwise optimal conditions, but in practice conditions make turnover a lot longer particularly in high latitudes.

Line 114: "verifying" is perhaps an overly strong word here given inherent uncertainties on your diagnosed τ_i , perhaps "supporting" instead.

Line 118-121: presumably this isn't because the ESM results are showing the enviro-constrained rather than intrinsic turnover times? And any thoughts as to why this is the case, both the general offset and the places where it's greatest? (Limitations to ESM treatment of soil carbon dynamics is outlined in introductory paragraph, but a little bit here on key drivers of this specific tendency would be useful for understanding ESM limitations and improvements in them could be focused in future.)

Line 128: as per line 47, does "compact" here also mean reduced complexity? It's not a term for models I'm so familiar with. Also, while it's clear that this is a new model being introduced, it would help to briefly highlight within the main text why a new model is needed to make this projection and how it relates to and builds on the previous model (given the previous RC model is also capable of making soil carbon stock projections in Fig S6, but presumably is limited to simplistic ESM soil carbon dynamics so needs extra aspects re. turnover times you've identified as important added). Fine to leave details to methods/SI some context would improve the logical flow here.

Line 152: there are more than three emission scenarios, so can just say "under three different emissions scenarios" (removing "the" for clarity)

Line 156-157: as phrased "will become a large carbon source under SSP1-2.6 (-19.1 PgC) and lose 24 and 45.2 Pg of carbon under SSP2-4.5 and SSP5-8.5..." makes it sound like source and loss are different things.

Line 157-159: bit of a confusing sentence – is the first sentence half reiterating the previous sentence

that the constrained simulations show a switch to source? Consider rephrasing / merging second half with above sentence if so, because as phrased it sounds "the constraint" is some kind of within-model event triggering the switch.

Line 160: I think why this is expected needs a bit of brief clarification – presumably because faster intrinsic turnover means soil carbon stocks react and degrade faster with warming (but possibly with caveat of the apparent/intrinsic turnover time mismatch reducing this a bit in practice?)

Line 162-163: "largely weakening the role of ecosystems in carbon sequestration potential in the future" reads a bit unclear to me – could rephrase as e.g. "reducing the potential capacity of land biosphere carbon sequestration in future". Also, which scenario does the 59 PgC increase plant biomass come from in the previous sentence (so that it can be more directly compared to your scenario-dependent soil loss estimates)?

Line 176-177: CO₂ fertilisation may be over-estimated in some ESMs, e.g. Wang et al., 2020 (<https://science.sciencemag.org/content/370/6522/1295>) or Winkler et al. 2021 (<https://bg.copernicus.org/articles/18/4985/2021/>), which combined with less soil carbon uptake than expected might feasibly be enough to get a net land carbon source under some scenarios.

Line 181-182: whenever a "we" is invoked in climate mitigation it should be clear who the we is referring to – e.g. is this humanity as a whole, high emitting countries/demographics, policymakers...

Line 187-188: it would also be interesting to see this done for 1.5C, given that's a goal much discussed in climate policy – would that be possible in your analysis as well?

Line 195: maybe "indicates" instead of "showed", as the model is only an estimate of course.

Dr. David A. McKay

Reviewer #2 (Remarks to the Author):

The authors fit fluxes from soil incubation data to a simple three-pool decomposition model to constrain estimates of carbon turnover times. The data are gathered from a large number of independent studies over global range of biomes, soil types and temperatures. The turnover times are then upscaled using machine learning methods. They compare the constrained turnover times to those in Earth System models (ESMs) in CMIP6 that have a similar three-pool representation of soil carbon and find that the majority of these models overestimate C turnover times compared to the incubation-derived estimates. A compact version of the model, also containing simple representations of priming and physical-chemical protection mechanisms, is constrained with additional global data sources. By forcing this model with ESM-predicted temperature and productivity, the authors find that there is a stronger positive feedback from soil carbon with warming than predicted by current ESMs, potentially making climate mitigation

efforts more challenging.

I find this study interesting and very relevant as it synthesizes many datasets together to constrain a critical feedback in ESMs. but have a few questions on methodology. It's an important and robust conclusion that the models overestimate turnover times and that has impacts on carbon cycle responses. But perhaps more important here is that the derived temperature sensitivities are quite high and this very likely plays a key role in changing the sign of the soil carbon response with warming. This appears in supplementary table 8 - the Q10 values for the slow and passive pools are substantially larger than used in most ESMs (2.85 and 3.77). However it's harder to judge the robustness of this result given the methods. Can the authors more quantitatively assess the relative roles of model bias in base turnover time compared to bias in temperature sensitivity?

Specific comments:

Lines 98-99: "Our estimate of the global mean (316 yr) is more than 16 times shorter than the radiocarbon-derived estimate..." – why not compare the estimated carbon age directly?

Lines 107-108: The ESMs describe pool-specific turnover times in their literature (e.g. Koven et al. 2013). However they may not be directly comparable as they in some cases have aggregated more than three pools into the three reported (Cfast, Cslow, Cpassive), and the model structure may be substantially different than used in this study.

Lines 108-109: What accounts for the spatial differences in estimated intrinsic turnover times in the models? In most of these models the intrinsic turnover times (usually at 20 or 25C under ideal moisture) are fixed parameters. The spatial variation then must be caused by differences in structure or parameters between your model and the decomposition module in the ESM. For example, the Q10 value in CESM is 1.5, whereas the assumed value here is significantly higher (Fig. S16). Therefore the derived turnover time in the tropics at the base temperature of 15C is higher, but this could be an artifact of the difference in temperature functions.

Lines 134-135: How sensitive is the result to the priming effect? As mentioned most ESMs don't estimate priming. By including priming in the compact model, the turnover times will decrease because of increasing root respiration, but a possible feedback in the ESM of increased nutrient availability for plant growth (increased NPP) would be missing.

Lines 378-388: Why not fit Q10, or some function that predicts Q10 as a function of temperature rather than using an assumed relationship? The large coverage of temperatures in the incubations (4 to 35C) should allow for a good fit.

Lines 435-6 (equation 8): Does this mean total NPP is added to the Cf pool each year? If NPP is increasing over time due to CO2/warming, litter production will lag NPP due to vegetation turnover times (especially for woody vegetation). Would it be better to use litter production, or is this variable not

available?

Lines 445-6: Here a constant $Q_{10} = 2.5$ is used, but for consistency should it not be the same relationship plotted in Fig. S16?

Lines 548-562 (sensitivity tests): Please describe these in more detail. In experiment 1, the Arrhenius function is used to scale the turnover times to 15C. Were all of the parameters then recalibrated, including the Q_{10} for the different pools? Would it make sense to add another sensitivity experiment where you use the Q_{10} values as prescribed in the models to test the impact of temperature sensitivity compared to base turnover rates?

To Reviewer #1

General Comments

[Comment 1] *In this paper, the authors present a compilation of observational data of soil carbon turnover times in different biomes and climate zones. This analysis shows that the intrinsic turnover time (i.e. turnover if the soil were at optimum environmental conditions) is highest in cold / high latitude regions, and lowest at warmer low latitudes. The seeming discrepancy with observed slow carbon degradation in high latitudes soils is because the apparent turnover time differs from the intrinsic turnover time due to non-optimum environmental conditions (so if high-latitude soil were subjected to more optimal conditions, the apparent turnover time would converge with the intrinsic turnover time). This data is then used to develop a simple model of soil carbon pools to estimate global patterns of intrinsic turnover time, the results of which compare favourably to estimates derived from a reduced complexity model emulating the soil carbon dynamics of Earth System Models. However, while the spatial patterns align the ESMs substantially over-estimate the magnitudes of intrinsic turnover time. To show what impact this has on future soil carbon projections, another model is developed by calibrating the previous reduced complexity model with intrinsic turnover time observations and adding additional processes (mineral protection and rhizosphere priming) and then run to 2100 under low/medium/high emissions scenarios. This indicates soil carbon will switch from a carbon sink to source under all scenarios, offsetting some of the carbon sequestered in plant biomass and so reducing the carbon budget for keeping global warming*

below 2C.

[Response] Thank you so much for your meticulous review, and valuable comments and suggestions on our manuscript. The point-by-point responses are listed following each comment/suggestion.

[Comment 2] *In this review I am focusing on the analysis logic and Earth system implications rather than the methodological details, which I'm not so familiar with for this field but appears to be sound, supports the conclusions made, and is likely reproducible from the data provided (although uploading code to a repository e.g. github/zenodo as well would improve reproducibility further).*

[Response] We thank the reviewer for the positive feedback. Following your suggestion, we have uploaded data and code in this study to the Figshare data repository (<https://figshare.com/s/0febe56304920b8536f0>).

[Comment 3] *In general this is an interesting and useful paper with clear relevance to Earth system modellers and future climate policy, and should be published with some text revisions to clarify some things. It is well known that soil carbon dynamics is one of the most uncertain aspects of Earth system models, and this study helps to illustrate and quantify the gap between ESMs and observations, indicating where ESMs can improve in future. However, there's quite a lot packed in to this paper (with results from multiple data analyses and models included), so there's not a lot of space to explain things fully and put them in context for readers who are less familiar with*

soil carbon turnover and its implications. For example, I think the difference between the intrinsic and apparent turnover times should be unpacked and explained more, as it's not necessarily obvious to non-specialist readers and is a crucial point for understanding the results of this paper. Additional explanation on model rationale and why ESMs are over-estimating turnover time and thus carbon sequestration would be helpful too, with the latter also important for outlining the implications of where ESMs could focus future improvements (which could be explicitly noted later too). Word count is no doubt a limitation here, but I think some brief extra sentences providing context where flagged may be sufficient to resolve this (otherwise a longer paper would be better to do this work justice). Finally, calculating the implications for the 1.5C carbon budget as well as 2C would no doubt be of interest to many. Specific details on where these points apply are below.

[Response] Thank you for your positive feedback and valuable suggestions.

Following your kind suggestions and comments, we have conducted a throughout revision of the manuscript. First, we have provided a detailed explanation of the difference between intrinsic and apparent carbon turnover times in both the Method section and the main text (see detailed responses to **Comment 12** by **Reviewer #1**). Second, we have explicitly discussed the possible reasons for the relatively short intrinsic turnover time in ESMs and outlined potential directions for future modelling efforts (see detailed responses to **Comment 15** by **Reviewer #1**). Third, we have added an additional analysis for the 1.5 °C temperature target (see detailed responses to **Comment 5** by **Reviewer #1**).

Specific Comments

[Comment 4] *Line 23-27: Does “after observational constraints” mean from now, or later on? I suggest rephrasing this sentence a bit to make this clearer (could also consider splitting the sentence after “worst emission scenarios”).*

[Response] Thanks for your suggestion. In revised manuscript, we have rephrased this sentence to make it clearer as follows: *“Our constraint showed that the global soils will switch from carbon sink to source, with a loss of 0.22–0.53 petagrams of carbon per year until the end of this century from strong mitigation to worst emission scenarios, suggesting that global soils will provide a strong positive carbon feedback on warming.”* (Lines 21-25 on Page 2)

[Comment 5] *Line 29-29: As per the more detailed comment on this below, what about implications for the carbon budget for 1.5C?*

[Response] Following your kind suggestion, we have also calculated the remaining carbon budget for limiting global warming well below 1.5 °C. Our results showed that global soils will sequester 45 (33–58) PgC less than the estimate based on the original Earth system model projections by 2100 for the 1.5 °C warming target (Figure R1). Based on this result, the remaining carbon budget, which is currently estimated at 110 PgC, should be further reduced by 41% over the course of this century to achieve the 1.5 °C warming target. This information has been added in the revised manuscript (Lines 197-207 on Page 11).

Figure R1 (also shown as Figure S11 in revised manuscript). Relationship between the magnitude of global warming and observational-constrained changes in global soil carbon stock across three different emission scenarios and five models.

[Comment 6] Line 32: A minor point, but might be clearer language to say something like “may act as a positive feedback on climate change” rather than “would act...”

[Response] Done as suggested.

[Comment 7] Line 33: I found “in the real Earth system” to be oddly phrased on first reading (as those experiments in the previous sentence were done on real parts of the Earth system too, albeit placed under artificial incubation conditions), though I do know that you mean across the whole Earth system in practice rather than in a few experiments here.

[Response] In revised manuscript, we have deleted this odd expression.

[Comment 8] *Line 35: I think it'd be useful either here or somewhere early on to briefly define and explain soil carbon turnover for less familiar readers, especially as this is not a disciplinary journal.*

[Response] Thanks for your suggestion. We have added a sentence to explain soil carbon turnover time as follows: *“One significant, and poorly understood, component of the system is the soil carbon turnover^{1,6}, which is defined as the average time it takes for a carbon atom to enter and leave the soil system⁷.”* (Lines 33-35 on Page 3)

[Comment 9] *Line 47: what do you mean by a “compact” model? Either brief explanation of that later on, or if you mean reduced complexity, or simple, or stylised model then that that would be more familiar phrasing (at least to me).*

[Response] Following your kind suggestions, we have discarded the use of “a compact model” and used *“refined reduced-complexity model”* throughout the manuscript.

[Comment 10] *Line 53: while the dataset gathered is pretty comprehensive across different biomes, there are some limitations (e.g. not so many datasets for non-forests, and not many sites in Africa or Central/South/Southeast Asia). I'm sure this reflects data availability and so is somewhat inevitable, but it could bear mentioning as a limitation in at least the methods section (as sometimes ecosystems behave a bit differently on different continents despite being in the same global biome). Also, in the*

map there are a few North European sites in e.g. the UK & North Germany that appear to be labelled as Boreal Forest when by most classifications they'd be in the Temperate biome (unless they're labelled as Boreal because they're from conifer plantations, but that would lead to different limitations) so this might need checking, especially as this would reduce Palearctic Boreal representation. Lastly on Fig. 1, at first glance it'd be easy to assume the colours in panels b-d are meant to match the biome colours above, so for clarity might be preferable to make as different shades of a new colour if possible, or to maintain continuity with Fig. 2 re-do the 1a dots as different symbols instead (the red wetland dots in panel a may also clash with green for colour-blind readers).

[Response] Thanks so much for your understanding and valuable suggestions.

Although our data set encompassed most of environmental conditions on the Earth (Figure S15 in revised manuscript), tropical regions such as Africa, high latitude areas and central and southern Asia are still underrepresented. Following your suggestion, we have discussed this limitation in both main text and Method section of the revised manuscript as follows:

“By including more data, particularly from under-sampled regions, such as Africa, central and southern Asia and some high latitudes, similar constrained projection studies are likely to provide further value to this area of research.” (Lines 216-219 on Page 12)

“Despite this, certain regions (such as Africa, central and southern Asia and some high latitudes) are underrepresented by our samples (Fig. 1). Thus, more long-term

soil incubations are urgently needed in these specific regions.” (Lines 343-345 on Page 18)

In addition, we have also carefully checked the information on forest type for each North European site (Table R1), and found that most of forest types at these sites were correctly categorized based on the tree species in this study. As the reviewer correctly pinpointed, there are two sites from the United Kingdom that are classified as planted forests, while the obtained soil τ_i patterns across different biomes are robust even when these two sites were moved into the “temperate forest” category (Figure R2). We also fine-tuned the color and shape of the biome markers to ensure better differentiation (Figure R3). All this information has been added in the method section of the revised manuscript.

Table R1. Boreal forest site characteristics across the North Europe.

Site	Latitude (°N)	Longitude (°E)	Country	Species	Type	Reference
Galloway	55.1	-4.5	UK	Sitka spruce	Planted (30yr)	Neale et al., 1997
Tharandt	50.97	13.57	Germany	Picea abies	Natural	Rey et al., 2006
Wetzstein	50.54	11.45	Germany	Picea abies	Natural	
Harwood	55.2	2.03	UK	Picea sitchensis	Planted (40yr)	
Loobos	52.17	5.73	Netherlands	Pinus sylvestris	Natural	

Figure R2. Boxplots showing the distributions of soil τ_i of C_{fast} (a), C_{slow} (b) and $C_{passive}$ (c) that are inverted from all experiments within each of the eight biomes, respectively. Specially, for the “temperate forest” category, we added data of two planted forests from UK.

Figure R3 (also shown as Figure 1 in revised manuscript). Distribution of intrinsic soil carbon turnover times from soil incubation experiments.

[Comment 11] *Line 70: could briefly mention a headline quantification of the importance of MAT here from Fig S1 (looks like c. 27-42% across the different pools) to underline this point.*

[Response] Thanks for your suggestion. We have rewritten this sentence as follows:

“Of the tested predictors, mean annual temperature (MAT) was the most important variable in explaining cross-site variability of soil τ_i for all three soil carbon pools, with relative variable importance of 27-42% across the different pools (Supplementary Fig. 1).” (Lines 76-79 on Page 5)

[Comment 12] *Line 73-77: the optimal environmental conditions caveat is key here, as a non-familiar reader might be otherwise surprised that colder regions are associated with faster soil carbon turnover given long-term C storage of say permafrost. I feel like this could be made clearer upfront to avoid confusion.*

[Response] Thanks for your constructive suggestions. In order to clarify the difference between intrinsic and apparent soil carbon turnover times, we have firstly added a brief explanation in the introduction of main text:

“Notably, soil τ_i is representative of the theoretical carbon turnover time under optimal conditions. While various environmental constraints such as freezing and physical protection could inhibit the achievement of this theoretical value, leading to longer apparent value of τ_i in the real-world settings⁸ (Methods).” (Lines 47-50 on Pages 3-4)

Then, in the Method section, we have added detailed explanations regarding the difference between intrinsic and apparent carbon turnover times.

“The intrinsic soil carbon turnover time reflects kinetic properties of various soil organic compounds under optimal conditions⁸. Generally, molecular structures with greater complexity exhibit an increased resistance to decomposition³⁶, resulting in a longer τ_i . However, in real-world conditions, various environmental constraints, including freezing, flooding and physical protection, can dampen decomposition processes⁸, frequently leading to a longer apparent value of τ_i .” (Lines 223-228 on Page 12)

[Comment 13] *Line 100-103: again, I think this could do with highlighting/a bit more explanation to make it clear to less-familiar readers that intrinsic turnover time is what's expected in theory given otherwise optimal conditions, but in practice conditions make turnover a lot longer particularly in high latitudes.*

[Response] Thanks for your suggestion. Please see detailed responses to Comment 12.

[Comment 14] *Line 114: “verifying” is perhaps an overly strong word here given inherent uncertainties on your diagnosed τ_i , perhaps “supporting” instead.*

[Response] Thanks for your suggestion. Following the Reviewer #2’s suggestion (**Comment 5** by **Reviewer #2**), we have removed this paragraph in the revised version.

[Comment 15] *Line 118-121: presumably this isn't because the ESM results are showing the enviro-constrained rather than intrinsic turnover times? And any thoughts as to why this is the case, both the general offset and the places where it's greatest? (Limitations to ESM treatment of soil carbon dynamics is outlined in introductory paragraph, but a little bit here on key drivers of this specific tendency would be useful for understanding ESM limitations and improvements in them could be focused in future.)*

[Response] Thanks for your suggestion. Following your constructive suggestion, we have added the following sentences to explain the results: *“The model-data bias of soil τ_i could be attributed to the omission of critical microbial processes from ESMs, such as thermal adaptation²⁴. Specifically, microbial turnover rates have been shown to adjust to temperature changes via biochemical trade-offs in enzyme and cell membrane structure and function^{25,26}. Low temperatures typically select for enzymes and/or membranes that are highly flexible to efficiently alter conformation and facilitate interactions²⁷. As a result, cold-adapted microbial communities have faster growth and respiration rates than the warm-adapted when compared at common temperatures^{5,26}.”* (Lines 121-128 on Pages 7)

In addition, after carefully examining each ESM parametrization, we realized that intrinsic turnover times in different carbon pools from all ESMs are fixed parameters without any spatial variability. In the revised manuscript, we therefore removed the part related to the spatial diagnosis of intrinsic turnover times from ESMs, and

directly compared data-driven intrinsic turnover times to those prescribed in ESMs. Our updated results showed that global soil τ_i is still overestimated by models (~30% across carbon pools). We have therefore updated figures and modified the text correspondingly in the revised manuscript (see detailed responses to **Comment 5** by **Reviewer #2**).

[Comment 16] *Line 128: as per line 47, does "compact" here also mean reduced complexity? It's not a term for models I'm so familiar with. Also, while it's clear that this is a new model being introduced, it would help to briefly highlight within the main text why a new model is needed to make this projection and how it relates to and builds on the previous model (given the previous RC model is also capable of making soil carbon stock projections in Fig S6, but presumably is limited to simplistic ESM soil carbon dynamics so needs extra aspects re. turnover times you've identified as important added). Fine to leave details to methods/SI some context would improve the logical flow here.*

[Response] Thanks for your suggestion. In this study, we developed a reduced-complexity three-pool model to approximate soil carbon dynamics in complex ESMs, and then refined this reduced-complexity model by including representations of priming effect and physical protection to constrain soil carbon projections.

To well resolve the reviewer's concern, we added "*Although our constructed reduced-complexity model is capable of mimicking ESMs soil carbon dynamics and making projections, it is limited to rudimentary processes without integrating*

emerging knowledge of controls on soil carbon turnover time.” into the revised manuscript (lines 134-137 on Page 8).

In addition, we have also added the following text into the Method section: *“Our constructed reduced-complexity model was demonstrated to well mimic soil carbon dynamics in ESMs (Supplementary Fig. 6), but it generally assumes that decomposition rates are only constrained by temperature and moisture availability⁹⁻¹¹. In fact, emerging processes, such as mineral protection and the rhizosphere priming effect, have the potential to affect soil carbon turnover time⁸, but which were mostly absent in current ESMs. We therefore refined the reduced-complexity model by including the impacts of climatic factors and these emerging processes (equation (11–12)) on soil carbon turnover time.”* (Lines 396-403 on Pages 20-21)

[Comment 17] *Line 152: there are more than three emission scenarios, so can just say “under three different emissions scenarios” (removing “the” for clarity)*

[Response] Done as suggested.

[Comment 18] *Line 156-157: as phrased “will become a large carbon source under SSP1-2.6 (-19.1 PgC) and lose 24 and 45.2 Pg of carbon under SSP2-4.5 and SSP5-8.5...” makes it sound like source and loss are different things.*

Line 157-159: bit of a confusing sentence – is the first sentence half reiterating the previous sentence that the constrained simulations show a switch to source? Consider rephrasing / merging second half with above sentence if so, because as phrased it

sounds "the constraint" is some kind of within-model event triggering the switch.

[Response] Following your kind suggestion, we have rewritten these two sentences as follows: *“In contrast, constrained ESM simulations showed that global soils will lose 19.1 Pg of carbon under SSP1-2.6, and this loss will be further escalated to 24 and 45.2 PgC under SSP2-4.5 and SSP5-8.5, respectively (Fig. 3). These projections translate into an estimated annual soil carbon loss rate of 0.22–0.53 PgC from now until the end of this century under different emissions scenarios.”* (Lines 157-161 on Page 9)

[Comment 19] *Line 160: I think why this is expected needs a bit of brief clarification – presumably because faster intrinsic turnover means soil carbon stocks react and degrade faster with warming (but possibly with caveat of the apparent/intrinsic turnover time mismatch reducing this a bit in practice?)*

[Response] Thanks for your suggestion. We have rephrased this sentence as follows. Furthermore, we acknowledge that environmental constraints could potentially obscure this effect in practice. However, our sensitivity analysis based on the reduced model confirmed the robustness of this interpretation (Sensitivity Experiment 3; Figure S19). *“This finding is generally consistent with the expectation that the intrinsic turnover time was overestimated in complex ESMs (Fig. 2). This is because faster intrinsic soil carbon turnover would lead to a more rapid decline in soil carbon stock in response to warming, although environmental constraints such as climatic limitation and physical protection would inhibit the achievement of this intrinsic*

*value*⁸.” (Lines 161-166 on Page 9)

[Comment 20] *Line 162-163: “largely weakening the role of ecosystems in carbon sequestration potential in the future” reads a bit unclear to me – could rephrase as e.g. “reducing the potential capacity of land biosphere carbon sequestration in future”. Also, which scenario does the 59 PgC increase plant biomass come from in the previous sentence (so that it can be more directly compared to your scenario-dependent soil loss estimates)?*

[Response] Thanks for your suggestion. We have rewritten this sentence as follows. *“Such soil carbon losses could largely offset increases in plant biomass due to the fertilization effect of rising CO₂, thereby reducing the potential capacity of land biosphere carbon sequestration in future. For example, in an intermediate emission scenario, the increase in CO₂ expected by 2100 would enhance the global plant biomass by 59 PgC using an empirical upscaling of CO₂ fertilization effect. However, this enhancement in biomass carbon stock would be substantially counterbalanced by soil carbon losses (24 PgC).”* (Lines 166-171 on Page 9)

[Comment 21] *Line 176-177: CO₂ fertilisation may be over-estimated in some ESMs, e.g. Wang et al., 2020 (<https://science.sciencemag.org/content/370/6522/1295>) or Winkler et al. 2021 (<https://bg.copernicus.org/articles/18/4985/2021/>), which combined with less soil carbon uptake than expected might feasibly be enough to get a net land carbon source under some scenarios.*

[Response] In revised manuscript, we have incorporated this point and rewritten the sentence as follows: *“However, the magnitude of soil carbon losses from tropical forests under SSP5-8.5 (21.4 PgC) is lower than a recent estimate (65 PgC) that is simply extrapolated from a two-year warming experiment in a tropical forest under 4°C warming³. The relatively low soil carbon loss is also due to the large increase in soil carbon input due to CO₂ fertilization in ESMs. While emerging evidence suggests that this CO₂ fertilization effect in ESMs may have been overestimated³³, and our estimates of soil carbon losses are then likely underestimated.”* (Lines 184-190 on Page 10)

[Comment 22] Line 181-182: whenever a "we" is invoked in climate mitigation it should be clear who the we is referring to – e.g. is this humanity as a whole, high emitting countries/demographics, policymakers...

[Response] Following your kind suggestion, we have rephrased this sentence as *“The vanished capacity of soils to sequester CO₂ suggests that a more aggressive strategy towards emissions reduction is required to realize the pledges of the Paris Agreement.”* (Lines 193-194 on Page 10)

[Comment 23] Line 187-188: it would also be interesting to see this done for 1.5C, given that's a goal much discussed in climate policy – would that be possible in your analysis as well?

[Response] Following your kind suggestion, we have also calculated the remaining

carbon budget for limiting global warming well below 1.5 °C (please see detailed responses to **Comment 5** by *Reviewer #1*).

[Comment 24] *Line 195: maybe "indicates" instead of "showed", as the model is only an estimate of course.*

[Response] Done as suggested.

To Reviewer #2

General Comments

[Comment 1] *The authors fit fluxes from soil incubation data to a simple three-pool decomposition model to constrain estimates of carbon turnover times. The data are gathered from a large number of independent studies over global range of biomes, soil types and temperatures. The turnover times are then upscaled using machine learning methods. They compare the constrained turnover times to those in Earth System models (ESMs) in CMIP6 that have a similar three-pool representation of soil carbon and find that the majority of these models overestimate C turnover times compared to the incubation-derived estimates. A compact version of the model, also containing simple representations of priming and physical-chemical protection mechanisms, is constrained with additional global data sources. By forcing this model with ESM-predicted temperature and productivity, the authors find that there is a stronger positive feedback from soil carbon with warming than predicted by current ESMs, potentially making climate mitigation efforts more challenging.*

[Response] Thank you so much for your valuable comments/suggestions on our manuscript. The point-by-point responses are listed following each comment/suggestion.

[Comment 2] *I find this study interesting and very relevant as it synthesizes many datasets together to constrain a critical feedback in ESMs. but have a few questions on methodology. It's an important and robust conclusion that the models overestimate*

turnover times and that has impacts on carbon cycle responses. But perhaps more important here is that the derived temperature sensitivities are quite high and this very likely plays a key role in changing the sign of the soil carbon response with warming. This appears in supplementary table 8 - the Q_{10} values for the slow and passive pools are substantially larger than used in most ESMs (2.85 and 3.77). However it's harder to judge the robustness of this result given the methods. Can the authors more quantitatively assess the relative roles of model bias in base turnover time compared to bias in temperature sensitivity?

[Response] Thanks for your constructive suggestions. To resolve your concern regarding the relative importance of soil τ_i and Q_{10} in soil carbon projections, we have added a sensitivity experiment in the revised manuscript (Figure R4). In this sensitivity experiment, we performed the three tests using the reduced-complexity model, in which input variables such as net primary productivity, mean annual temperature and mean annual precipitation were derived from original ESMs. Specially, in the first test, we replaced soil τ_i with our data-driven estimates, and replaced Q_{10} for each pool with that derived from the refined reduced-complexity model, in which Q_{10} for each pool was obtained through calibration against observed soil carbon stocks (Table S9). For the second test, we replaced soil τ_i with our data-driven estimates and used ESM's own Q_{10} and associated reference temperatures (Table R2). For the third test, we used ESM's own soil τ_i , but assigned Q_{10} for each pool to the calibrated one.

Our results showed that global soils will become a source of carbon to the

atmosphere by the end of this century under different emissions scenarios. However, compared to the Q_{10} -constrained model (13 PgC averaged across scenarios), the magnitude of soil carbon loss in the τ_i -constrained model (19.5 PgC) is much closer to that in model constrained by the both (26.3 PgC), suggesting that soil τ_i is more important than Q_{10} in determining projections of soil carbon dynamics. This information has been added in the revised manuscript (see detailed responses to **Comment 10** by *Reviewer #2*).

Figure R4. Projected changes in global soil carbon stocks from original ESMs, reduced-complexity (RC) model, RC model constrained by soil τ_i , Q_{10} and the both.

Table R2 (also shown as Table S2 in revised manuscript). Model-specified soil intrinsic turnover times (τ_i ; yr) of different carbon pools, Q_{10} and associated reference temperature ($^{\circ}\text{C}$).

Model	Soil fast	Soil slow	Soil passive	Q_{10}	T_{ref}	Reference
	τ_i	τ_i	τ_i			
ACCESS-ESM1-5	0.24	5	222	1.72	20	Rafique et al., 2016
CESM2-WACCM	0.17	6.1	270	1.5	25	Koven et al., 2013
IPSL-6A-LR	0.149	5.48	241	2	30	Krinner, et al., 2005
NorESM2-LM	0.17	6.1	270	1.5	25	Koven et al., 2013
TaiESM1	0.17	6.1	270	1.5	25	Koven et al., 2013

Specific Comments

[Comment 3] *Lines 98-99: “Our estimate of the global mean (316 yr) is more than 16 times shorter than the radiocarbon-derived estimate...” – why not compare the estimated carbon age directly?*

[Response] In our study, the global estimates of soil carbon turnover times are intrinsic ones upscaled from the incubation experiments, rather than apparent or realized turnover time that can be approximated by radiocarbon-derived estimate. We have rewritten this sentence to make it clearer as follows:

“To explore the extent to which the intrinsic turnover times translate into the realized ones due to environmental constraints⁸, we compared our estimates with the

radiocarbon-derived carbon age as a surrogate of realized or apparent turnover times (Supplementary Table 1).” (Lines 103-106 on Page 6)

[Comment 4] *Lines 107-108: The ESMs describe pool-specific turnover times in their literature (e.g. Koven et al. 2013). However they may not be directly comparable as they in some cases have aggregated more than three pools into the three reported (C_{fast} , C_{slow} , $C_{passive}$), and the model structure may be substantially different than used in this study.*

[Response] The current state-of-the-art earth system models (ESMs) simulated soil carbon dynamics using conceptual soil pools with spatial-invariant intrinsic turnover rates. In the revised manuscript, we realized that intrinsic turnover times in different carbon pools from all ESMs are fixed parameters without any spatial variability, and therefore directly evaluated these fixed intrinsic turnover times in ESMs using our data-driven ones (see detailed responses to **Comment 5** by *Reviewer #2*).

As the reviewer correctly pinpointed, ESMs differed in structuring of carbon pools. According to outputs of ESMs archived in CMIP6 repository, models differed in the number of soil carbon pools ranging from one (e.g., CanESM5) to at least five (e.g., CESM2-WACCM), and the majority of models lack any depth-related information. Only a few models such as CLM have a vertical discretization of carbon pools at different soil depths (Koven et al., 2013). In this study, we only selected ESMs with at least three carbon pools. Although ESMs differed in the number of carbon pools and their associated parameters (such as Q_{10} and intrinsic carbon

turnover time), parameterizations of climatic constraints on soil carbon turnover times is structurally similar among different ESMs.

Here we have demonstrated that the reduced-complexity model with three carbon pools could well capture soil carbon dynamics of different pools in each ESM. This also holds true for those ESMs with more than three carbon pools, with their litter or woody debris carbon pools (*cLitter* and *cCwd*) being integrated with *cFast* (see also He et al., 2016). This result highlighted that using a reduced-complexity model framework to simulate complex ESMs is robust in capturing soil carbon dynamics, even in the presence of variations in the number of carbon pools.

Last but not the least, the main purpose of this study is to extract the intrinsic soil carbon turnover time from soil incubation experiments to constrain realized soil turnover rates and then project soil carbon dynamics based on the reduced-complexity modeling framework. While, the intrinsic soil carbon turnover times derived from soil incubation experiments did not account for variations across different soil depths and easily-decomposable pools such as litter. When developing a reduced-complexity modeling framework to constrain soil carbon projections, we solely utilized climatic and NPP data from ESMs, without incorporating any model data related to soil carbon. Therefore, different structuring of soil carbon pools in ESMs would have marginal impacts on our constrained soil carbon dynamics.

[Comment 5] *Lines 108-109: What accounts for the spatial differences in estimated intrinsic turnover times in the models? In most of these models the intrinsic turnover*

times (usually at 20 or 25C under ideal moisture) are fixed parameters. The spatial variation then must be caused by differences in structure or parameters between your model and the decomposition module in the ESM. For example, the Q10 value in CESM is 1.5, whereas the assumed value here is significantly higher (Fig. S16). Therefore the derived turnover time in the tropics at the base temperature of 15C is higher, but this could be an artifact of the difference in temperature functions.

[Response] Thank you for the constructive comment. After carefully examining each ESM parametrization, we realized that intrinsic turnover times in different carbon pools from all ESMs are fixed parameters without any spatial variability. In the revised manuscript, we therefore removed the part related to the spatial diagnosis of intrinsic turnover times from ESMs, and directly evaluated these fixed intrinsic turnover times in ESMs using our data-driven ones (Figure R5). We have updated figures and modified the text correspondingly in the revised manuscript.

Figure R5 (also shown as Figure 2 in revised manuscript). Global distributions of intrinsic soil carbon turnover times. a, d, g, Observation-derived intrinsic soil carbon turnover times (τ_i) at 1m depth that are upscaled from 374 data points at 15 °C using the machine learning model for C_{fast} (a), C_{slow} (d) and $C_{passive}$ (g), respectively. b, e, h, Standard deviations of global soil τ_i estimates derived from different combinations of data sets. c, f, i, Latitudinal profiles of soil τ_i , aggregated at 0.5° latitudinal resolution. The black and red lines indicate observation-derived and CMIP6 model-averaged soil τ_i , respectively. The pink dashed lines are soil τ_i specified in each model. The red shading represents the standard deviation. The x-axis data in c, f, i are log-transformed.

[Comment 6] *Lines 134-135: How sensitive is the result to the priming effect? As mentioned most ESMs don't estimate priming. By including priming in the compact model, the turnover times will decrease because of increasing root respiration, but a possible feedback in the ESM of increased nutrient availability for plant growth (increased NPP) would be missing.*

[Response] Thanks for your constructive comment. By comparing paired simulations with and without the rhizosphere priming effect in our reduced-complexity model (Figure R4), the priming effect, without considering the feedbacks between nutrients and soil carbon dynamics, reduced soil carbon stocks by 11-19 PgC across different emissions scenarios. As the reviewer correctly pinpointed, the rhizosphere priming effect could also increase the release of soil nutrients, which would in turn stimulate plant growth (Brzostek et al., 2012; Drake et al., 2013) and thereby create a positive feedback loop that further decreases soil carbon turnover time. This implied that the soil carbon stock would be further reduced because of this possible feedback on soil carbon turnover time. But on the other hand, enhanced plant growth due to the rhizosphere priming effect would partly offset soil carbon losses due to enhanced soil carbon turnover rates. Therefore, the net effect due to this positive feedback on soil carbon stock changes might not be so large. While, fully resolving this question requires next generation of Earth system models that explicitly incorporate the rhizosphere priming effect within the coupled carbon-nitrogen cycle framework. In order to well resolve the reviewer's concern, we have added the following discussion into the Method section of the revised manuscript.

“The rhizosphere priming effect could also increase the release of soil nutrients, which would in turn stimulate plant growth⁵⁷ and thereby create a positive feedback loop that further decreases soil carbon turnover time. On the other hand, enhanced plant growth due to the rhizosphere priming effect might partly offset soil carbon losses due to enhanced soil carbon turnover rates. The net effect due to this positive feedback on soil carbon stock changes might not be so large. This effect was not included in our refined reduced-complexity models due to a general absence of carbon-nitrogen coupling processes in most of ESMs. To fully resolve this question, we require next generation of Earth system models that explicitly incorporate the rhizosphere priming effect within the coupled carbon-nitrogen cycle framework.”

(Lines 420-431 on Pages 21-22)

[Comment 7] *Lines 378-388: Why not fit Q_{10} , or some function that predicts Q_{10} as a function of temperature rather than using an assumed relationship? The large coverage of temperatures in the incubations (4 to 35C) should allow for a good fit.*

[Response] In our compiled dataset, the incubation temperatures could range from 4 °C to 35 °C but most experiments incubated soils at only one temperature, making it difficult to establish a relationship between Q_{10} and temperature. Therefore, we used the empirical Q_{10} -temperature function of Hamdi et al. (2013), which was derived from incubation experiments reporting Q_{10} values. To avoid potential confusion, we have made this point clearer in revised manuscript: *“Here, we assumed that Q_{10} varies with temperature, and their empirical function was derived from a previous*

synthesis analysis of Q_{10} and temperature from laboratory studies across various ecosystems at the global scale²³ (Supplementary Fig. 14).” (Lines 311-313 on Page 16)

[Comment 8] *Lines 435-6 (equation 8): Does this mean total NPP is added to the Cf pool each year? If NPP is increasing over time due to CO2/warming, litter production will lag NPP due to vegetation turnover times (especially for woody vegetation). Would it be better to use litter production, or is this variable not available?*

[Response] Thanks for your comment. We acknowledge that using litter production as soil carbon input in this equation would be better, but this specific variable is not available in ESMs. While our reduced-complexity model still well captured the soil carbon dynamics simulated in ESMs (Figure R6), tentatively suggesting that NPP can be used as a good proxy for soil carbon inputs.

Figure R6 (also shown as Figure S6 in revised manuscript). Comparison of different global soil carbon stock projections of the three pools under SSP1-2.6, SSP2-4.5 and SSP5-8.5 scenarios.

[Comment 9] Lines 445-6: Here a constant $Q_{10} = 2.5$ is used, but for consistency should it not be the same relationship plotted in Fig. S16?

[Response] In the revised version, we have used model-prescribed Q_{10} in the reduced-

complexity model to approximate soil carbon dynamics simulated by complex ESMs (please see **Comment 4** by *Reviewer #2*).

[Comment 10] *Lines 548-562 (sensitivity tests): Please describe these in more detail.*

In experiment 1, the Arrhenius function is used to scale the turnover times to 15C.

Were all of the parameters then recalibrated, including the Q_{10} for the different pools? Would it make sense to add another sensitivity experiment where you use the Q_{10} values as prescribed in the models to test the impact of temperature sensitivity compared to base turnover rates?

[Response] Thanks for your constructive suggestions. In sensitivity experiment 1, the model parameters including Q_{10} for different pools were recalibrated. We have rewritten this section as follows and added a table to provide more detailed information on the sensitivity experiments (Table R3). Moreover, following your suggestion, we have also added several sensitivity experiments using the reduced-complexity model to resolve your concern regarding the relative importance of soil τ_i and Q_{10} in soil carbon projections (see detailed response to **Comment 2** by *Reviewer #2*).

“Sensitivity analysis. To assess the robustness of our constrained results, we conducted the following sensitivity experiments (SE) (Supplementary Fig. 19; Supplementary Table 10). First, we used the Arrhenius function instead of the empirical Q_{10} -temperature relationship in our default simulation to scale site-level τ_i at their own incubation temperatures to that at 15 °C and then further obtained global

estimates of soil τ_i in a machine learning algorithm linking these τ_i to environmental variables across sites. We then prescribed τ_i using these data-driven estimates based on the Arrhenius function in refined reduced-complexity model to constrain soil carbon projections (SE1). Second, the use of laboratory incubation experiments, albeit with the length of the period longer than six months, would still have uncertainties in the quantification of τ_i of slow-cycling carbon pool especially $C_{passive}$. In SE2, we prescribed τ_i of $C_{passive}$ as the ensemble mean of ESM's own values rather than the data-driven estimates in our default simulation (Fig. 2).

Third, in SE3-SE6, we used the reduced-complexity model (equation 8), which only considered climate controls on soil carbon turnover time and used model inputs such as NPP, MAT and MAP directly from the original ESMs, instead of the refined model (Equation 11) to constrain soil carbon projections in ESMs. In order to evaluate the relative importance of soil τ_i and Q_{10} in soil carbon projections, we performed the following SE3-SE5 tests. For SE3, we replaced soil τ_i with our data-driven estimates, and replaced Q_{10} for each pool with that derived from the refined reduced-complexity model (Equation 11), in which Q_{10} for each pool was obtained through calibration against observed soil carbon stocks (Supplementary Table 9). For SE4, we replaced soil τ_i with our data-driven estimates and used ESM's own Q_{10} and associated reference temperatures. For SE5, we used ESM's own soil τ_i , but assigned Q_{10} for each pool to the calibrated one. The magnitude of soil carbon loss in SE4 (19.5 PgC averaged across scenarios) is much closer to that in SE3 (26.3 PgC) than in SE5 (13 PgC), suggesting that soil τ_i is more important than Q_{10} in determining

projections of soil carbon dynamics. In addition, since the use of laboratory incubation experiments would have uncertainties in the quantification of τ_i of $C_{passive}$, for SE6, we used the ensemble mean of ESM's own soil τ_i of $C_{passive}$, but assigned τ_i of C_{fast} and C_{slow} , and Q_{10} to be the same with those in SE4. The results of SE6 showed that global soils would be a source of carbon to the atmosphere (10 PgC), albeit at a lower magnitude than SE4.” (Lines 473-500 on Pages 24-25)

Table R3 (also shown as Table S10 in revised manuscript). Description of sensitivity experiments (SE) in this study.

Options	Global data-driven soil τ_i estimates		Model structure		Model inputs (NPP, MAT and MAP)		Q_{10}		τ_i of C_{fast} and C_{slow}		τ_i of $C_{passive}$	
	Using empirical Q_{10} -temperature function to scale τ_i to 15°C at the site level	Using Arrhenius function to scale τ_i to 15°C at the site level	Climate, physical protection and priming controls on soil carbon turnover	Climate controls on soil carbon turnover	Satellite observations	Original ESMs	Calibrated against observations	ESM's own values	Data-driven values	ESM's own values	Data-driven	ESM's own values
Default	√		√		√		√		√		√	
SE1		√	√		√		√		√		√	
SE2	√		√		√		√		√			√
SE3	√			√		√	√		√		√	
SE4	√			√		√		√	√		√	
SE5	√			√		√	√			√		√
SE6	√			√		√		√	√			√

References

- Brzostek E R, Greco A, Drake J E, et al. Root carbon inputs to the rhizosphere stimulate extracellular enzyme activity and increase nitrogen availability in temperate forest soils[J]. *Biogeochemistry*, 2013, 115: 65-76.
- Drake J E, Darby B A, Giasson M A, et al. Stoichiometry constrains microbial response to root exudation-insights from a model and a field experiment in a temperate forest[J]. *Biogeosciences*, 2013, 10(2): 821-838.
- Hamdi S, Moyano F, Sall S, et al. Synthesis analysis of the temperature sensitivity of soil respiration from laboratory studies in relation to incubation methods and soil conditions [J]. *Soil Biology and Biochemistry*, 2013, 58: 115-126.
- He Y, Trumbore S E, Torn M S, et al. Radiocarbon constraints imply reduced carbon uptake by soils during the 21st century [J]. *Science*, 2016, 353(6306): 1419-1424.
- Koven C D, Riley W J, Subin Z M, et al. The effect of vertically resolved soil biogeochemistry and alternate soil C and N models on C dynamics of CLM4 [J]. *Biogeosciences*, 2013, 10(11): 7109-7131.
- Neale S P, Shah Z, Adams W A. Changes in microbial biomass and nitrogen turnover in acidic organic soils following liming [J]. *Soil Biology and Biochemistry*, 1997, 29(9-10): 1463-1474.
- Rafique R, Xia J, Hararuk O, et al. Divergent predictions of carbon storage between two global land models: Attribution of the causes through traceability analysis [J]. *Earth System Dynamics*, 2016, 7(3): 649-658.

Rey A N A, Jarvis P. Modelling the effect of temperature on carbon mineralization rates across a network of European forest sites (FORCAST) [J]. *Global Change Biology*, 2006, 12(10): 1894-1908.

REVIEWERS' COMMENTS

Reviewer #1 (Remarks to the Author):

Second review of “Projected soil carbon loss with warming in constrained Earth system models” (NCOMMS-23-13941A) for Nature Communications

The authors have carefully revised their manuscript in response to the reviewer comments, and I believe the manuscript has improved as a result and is ready to be published with minimal further revisions.

The authors have sufficiently clarified key concepts like soil carbon turnover time and explained the seemingly counter-intuitive difference between intrinsic and apparent turnover times, as well as explaining the implications and wider context of their results. Additionally, the authors have better explained modelling approach, clarified limitations in the global distribution of observational data (noting patchiness in Africa, southern Asia, and high latitudes), implemented most of the minor edits requested, and have calculated the implications for the 1.5C carbon budget as suggested, leading to a very topical result. A minor point on the latter is that a recent update to remaining carbon budget (<https://www.nature.com/articles/s41558-023-01848-5>) indicate an even smaller carbon budget is now available, which if incorporated would make your percentage reductions even bigger (but it may make more sense to keep to 2020 baseline, noting budget is shrinking since).

Explanation has also been added on why ESMs over-estimate turnover times (bearing in mind the wider revisions in this section in response to reviewer 2, which seems to have been sufficiently resolved). I’m wondering though if as well as latitudinal differences whether anything can be drawn out from inter-pool differences, e.g. the ESM slow pool average being generally a worse fit than the fast/passive pools. Additionally, while this study does implicitly indicate the areas which require improvement in ESMs as stated in the end summary, if there’s space a brief explicit mention of key processes to prioritise for inclusion in future ESMs would be a valuable addition.

The basis of N Europe sites being classified as Boreal has also been clarified as being a latitudinal cut-off of 50oN (and those sites representing coniferous forest and not being consequential to the results). This doesn’t quite fit the classic biogeographic definition of boreal, but is a reasonable global approximation (fitting N America better than Eurasia) and is made clear in the Methods.

Beyond the above, my only remaining specific comments are: considering changing “showed that” on line 21 to “suggests that” (as models can only over indicate and project, not definitively predict), defining “ESM” on first mention (currently several mentions in on line 112), considering the same magnitude scale on Fig S5a as other panels (to make them visually inter-comparable, with difference in panel a likely noticeably paler), and capitalising Arctic on line 57.

Dr. David A. McKay

Reviewer #2 (Remarks to the Author):

The authors have provided a comprehensive response to both reviews and have addressed nearly all of my concerns with additional analysis and clarification. I appreciate the additional sensitivity analysis to address my concerns about the impact of temperature sensitivity on the analysis.

My only remaining concern upon seeing table R2 is that all 5 ESMS have quite similar turnover rates. All 5 models, I believe, heavily draw from the CENTURY model (the three variants that use variants of CLM from Koven et al. (2013), IPSL, and ACCESS-ESM that uses CASA-CNP with a similar structure. Other models that have a different number of pools likely derive from models other than CENTURY and so may behave differently. Are these 5 models representative of the broader CMIP6 ensemble in terms of the responses to the different emissions scenarios? A brief mention of CENTURY's influence may be useful in the discussion, especially in the context of how biome-specific or otherwise spatially varying turnover rates may improve predictions.

This reference may be useful:

Berardi, D, Brzostek, E, Blanc-Betes, E, et al. 21st-century biogeochemical modeling: Challenges for Century-based models and where do we go from here? GCB Bioenergy. 2020; 12: 774–788.

<https://doi.org/10.1111/gcbb.12730>

To Reviewer #1

Second review of “Projected soil carbon loss with warming in constrained Earth system models” (NCOMMS-23-13941A) for Nature Communications

[Comment 1] *The authors have carefully revised their manuscript in response to the reviewer comments, and I believe the manuscript has improved as a result and is ready to be published with minimal further revisions.*

[Response] Thank you so much for your valuable and encouraging comments/suggestions. The point-by-point responses are listed following each comment/suggestion.

[Comment 2] *The authors have sufficiently clarified key concepts like soil carbon turnover time and explained the seemingly counter-intuitive difference between intrinsic and apparent turnover times, as well as explaining the implications and wider context of their results. Additionally, the authors have better explained modelling approach, clarified limitations in the global distribution of observational data (noting patchiness in Africa, southern Asia, and high latitudes), implemented most of the minor edits requested, and have calculated the implications for the 1.5C carbon budget as suggested, leading to a very topical result. A minor point on the latter is that a recent update to remaining carbon budget (<https://www.nature.com/articles/s41558-023-01848-5>) indicate an even smaller carbon budget is now available, which if incorporated would make your percentage*

reductions even bigger (but it may make more sense to keep to 2020 baseline, noting budget is shrinking since).

[Response] Following your constructive suggestion, we have updated the remaining carbon budget for 1.5°C (68 PgC, 50% chance) and 2°C (327 PgC, 50% chance), and found that a reversal of global soil carbon balance would lead to a reduction of about 66% and 15% in the remaining carbon budget for limiting global warming well below 1.5°C and 2°C, respectively. This information has been updated in the revised version (L26-L28 on Page 2; L198-L206 on Page 11).

[Comment 3] *Explanation has also been added on why ESMs over-estimate turnover times (bearing in mind the wider revisions in this section in response to reviewer 2, which seems to have been sufficiently resolved). I'm wondering though if as well as latitudinal differences whether anything can be drawn out from inter-pool differences, e.g. the ESM slow pool average being generally a worse fit than the fast/passive pools. Additionally, while this study does implicitly indicate the areas which require improvement in ESMs as stated in the end summary, if there's space a brief explicit mention of key processes to prioritise for inclusion in future ESMs would be a valuable addition.*

[Response] Thanks for your constructive suggestion. In the revised manuscript, we have mentioned the inter-pool difference and added a brief and clear description about future modelling efforts as follows.

“Overall, the model ensemble overestimated soil τ_i by about 30% across the three

different pools globally, with larger biases in C_{slow} (58%) than those in C_{fast} (12%) and $C_{passive}$ (17%) (Fig. 2).” (L122-L124 on Page 7).

“Future modelling efforts should seek a spatial representation of soil intrinsic turnover parameters especially for C_{slow} , e.g. by incorporating microbial metrics (such as thermal adaptation²⁴, species composition²⁸) into ESMs to build confidence in predicting soil carbon–climate feedback.” (L130-L134 on Page 7).

[Comment 4] *The basis of N Europe sites being classified as Boreal has also been clarified as being a latitudinal cut-off of 50oN (and those sites representing coniferous forest and not being consequential to the results). This doesn't quite fit the classic biogeographic definition of boreal, but is a reasonable global approximation (fitting N America better than Eurasia) and is made clear in the Methods.*

[Response] We appreciate your positive comment.

[Comment 5] *Beyond the above, my only remaining specific comments are: considering changing “showed that” on line 21 to “suggests that” (as models can only over indicate and project, not definitively predict), defining “ESM” on first mention (currently several mentions in on line 112), considering the same magnitude scale on Fig S5a as other panels (to make them visually inter-comparable, with difference in panel a likely noticeably paler), and capitalising Arctic on line 57.*

[Response] Done as suggested.

To Reviewer #2

[Comment 1] *The authors have provided a comprehensive response to both reviews and have addressed nearly all of my concerns with additional analysis and clarification. I appreciate the additional sensitivity analysis to address my concerns about the impact of temperature sensitivity on the analysis.*

[Response] Thank you so much for your valuable suggestions, which were very helpful for us to improve the quality of the manuscript.

[Comment 2] *My only remaining concern upon seeing table R2 is that all 5 ESMS have quite similar turnover rates. All 5 models, I believe, heavily draw from the CENTURY model (the three variants that use variants of CLM from Koven et al. (2013), IPSL, and ACCESS-ESM that uses CASA-CNP with a similar structure. Other models that have a different number of pools likely derive from models other than CENTURY and so may behave differently. Are these 5 models representative of the broader CMIP6 ensemble in terms of the responses to the different emissions scenarios? A brief mention of CENTURY's influence may be useful in the discussion, especially in the context of how biome-specific or otherwise spatially varying turnover rates may improve predictions.*

This reference may be useful:

Berardi, D, Brzostek, E, Blanc-Betes, E, et al. 21st-century biogeochemical modeling:

Challenges for Century-based models and where do we go from here? GCB

Bioenergy. 2020; 12: 774–788. <https://doi.org/10.1111/gcbb.12730>

[Response] Thanks for your suggestions. We have provided a brief mention of CENTURY's influence as follows.

“Notably, the five models used in this study may not be representative of the broader CMIP6 ensemble because they draw heavily from the CENTURY model and have then similar structures (e.g., three different soil carbon pools)³⁷. Other CMIP models derived from models other than CENTURY may behave differently and deserve further exploration.” (L397-L401 on Page 20).